# Imitation Learning for Multi-turn LM Agents
# via On-policy Expert Corrections

Niklas Lauffer[† 1]   Xiang Deng[† 2]   Srivatsa Kundurthy[† 3]   Brad Kenstler[4]   Jeff Da[† 5]

## Abstract

A popular paradigm for training LM agents relies on *imitation learning*, fine-tuning on expert trajectories. However, we show that the off-policy nature of imitation learning for multi-turn LM agents suffers from the fundamental limitation known as *covariate shift*: as the student policy's behavior diverges from the expert's, it encounters states not present in the training data, reducing the effectiveness of fine-tuning. Taking inspiration from the classic DAgger algorithm, we propose a novel data generation methodology for addressing covariate shift for multi-turn LLM training. We introduce *on-policy expert corrections (OECs)*, partially on-policy data generated by starting rollouts with a student model and then switching to an expert model part way through the trajectory. We explore the effectiveness of our data generation technique in the domain of software engineering (SWE) tasks, a multi-turn setting where LLM agents must interact with a development environment to fix software bugs. Our experiments compare OEC data against various other on-policy and imitation learning approaches on SWE agent problems and train models using a common rejection sampling (i.e., using verified reward) combined with supervised fine-tuning technique. Experiments find that OEC trajectories show a relative 14% and 13% improvement over traditional imitation learning in the 7b and 32b setting, respectively, on SWE-bench verified. Our results demonstrate the need for combining expert demonstrations with on-policy data for effective multi-turn LM agent training.

[†]Work was done while at Scale AI. [1]UC Berkeley [2]NeoCognition [3]Cornell University [4]Scale AI [5]Inflection AI. Correspondence to: Niklas Lauffer <nlauffer@berkeley.edu>.

*Proceedings of the 43rd International Conference on Machine Learning*, Seoul, South Korea. PMLR 306, 2026. Copyright 2026 by the author(s).

## 1. Introduction

Imitation learning from expert demonstrations has emerged as a popular paradigm for training language models (LMs) on agentic tasks involving multiple turns, interactions with an environment, or tool-uses. This approach typically involves fine-tuning a model on multi-turn expert trajectories that are collected offline and has been successfully applied to tasks in software engineering (SWE) (Hou et al., 2023), tool calling (Anthropic, 2024; Chen et al., 2024), and reasoning (Chen et al., 2023; Cobbe et al., 2021). Rejection sampling is often used to further filter the expert trajectories before fine-tuning (Ouyang & et al., 2022; Yang et al., 2025; Chen et al., 2023), particularly in code generation and mathematical reasoning tasks where solutions can be automatically verified against test cases or formal specifications (Mirzadeh et al., 2024).

A key limitation of traditional imitation learning is its susceptibility to covariate shift (Ross & Bagnell, 2010), which occurs when the state distribution encountered during deployment diverges from that seen during training (i.e., the distribution induced by the expert). This is a common problem in the domain of *multi-turn* imitation learning since inaccuracies in a learned policy can compound over several turns, leading to a very different state in the environment than observed under the expert demonstrations. In the case of LM agents, covariate shift can occur in the state of the underlying environment as well as through the agent's own token output, since all previous turns (including tool calls and chain-of-thought) are typically used as input for the LM agent's next action. We conduct an analysis by embedding multi-turn LM agent trajectories into a continuous representation space and show that they do indeed suffer from increasingly worse covariate shift throughout a trajectory.

In response to the challenge of covariate shift in LM agent training, we introduce a novel multi-turn distillation approach inspired by DAgger (dataset aggregation) (Ross et al., 2011), a classic imitation learning algorithm designed to mitigate covariate shift in multi-turn settings. DAgger works by rolling out trajectories *on-policy* and training on the actions an expert would have taken on each step of that trajectory. Our data generation method, which we call *on-policy expert corrections (OECs)* adapts the principle behind DAgger to

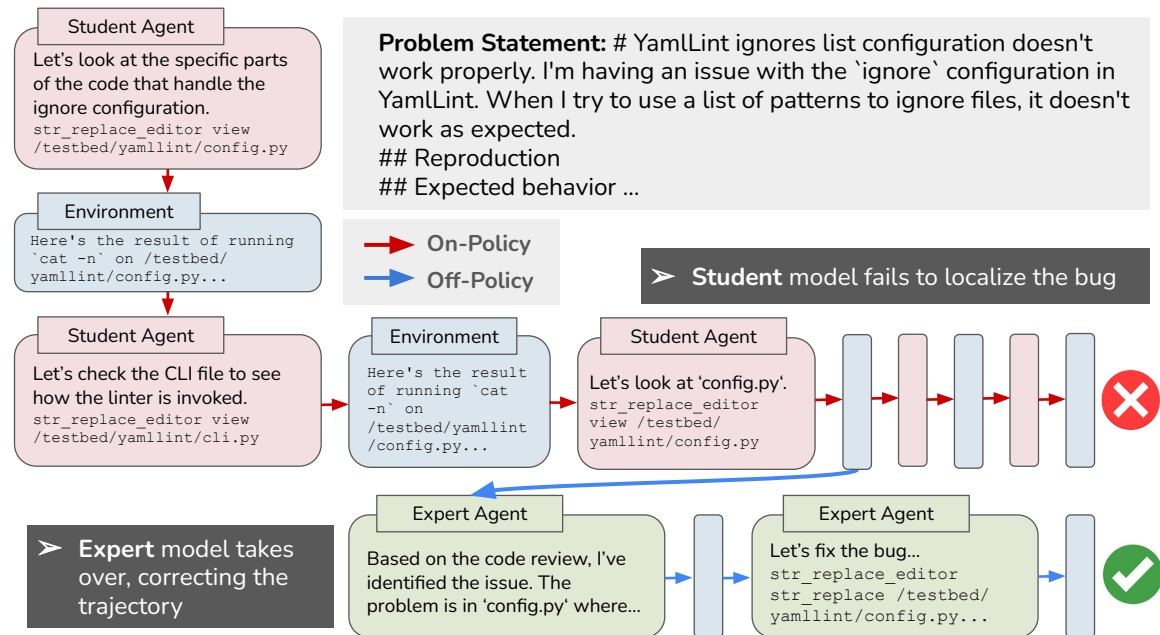

*Figure 1.* An example of a real on-policy expert correction in the SWE agent domain. The student begins the trajectory and fails to localize the bug but after the expert takes over, it brings the trajectory back on track to finish localization and write and submit a patch. The full transcript of this trajectory can be found in Section A.

LM agents by partially rolling out trajectories on-policy with the student model and then transitioning to the expert model to complete the trajectory. This approach allows us to collect partially on-policy data in an online way while retaining the benefits of training on expert data. Furthermore, unlike traditional DAgger, OECs allow the trajectory to be rolled out until completion, thereby allowing us to incorporate verifier reward (e.g., for rejection sampling). This approach effectively combines the strengths of on-policy approaches (such as reinforcement learning) with the benefit of imitation learning from expert data.

In order to study the efficacy of OEC, we explore training with several different data generation techniques in the SWE agent domain. SWE agent problems require an LM agent to interact with an isolated codebase containing a bug in order to write a code patch. Recently, fully agentic scaffolds (such as SWE-Agent (Yang et al., 2024) and OpenHands (Wang et al., 2025)) have emerged as the dominant system for deploying LM agents in SWE domains. In these fully agentic scaffolds, LMs must navigate the file system, read and write files, run unit tests, execute arbitrary bash commands, and finally submit a code patch. Depending on the complexity of the problem instance, SWE agent trajectories can take up to 80 turns to solve. Any substantial behavioral differences between a student and expert model can lead to covariate shift, including making mistakes such as localizing a bug to the wrong file, or differences in approaching the problem. Figure 1 depicts an example of a real OEC trajectory in a

SWE agent task. The student begins the trajectory and fails to localize the bug but after the expert takes over, it brings the trajectory back on track to finish localization and write and submit a patch.

We compare OECs against other popular data generation techniques: behavioral cloning and on-policy trajectories (comparable to reinforcement learning). We fine-tune Qwen2.5-Coder models using the popular rejection sampling (based on unit tests) with supervised fine-tuning formula. Our results demonstrate improved sample efficiency of OEC trajectories over both baselines and our ablations reveal other relevant insights about agentic LM training: (1) that unit test-based rejection sampling is not enough to guarantee high-quality training trajectories and (2) that fine-tuning on on-policy trajectories (even after filtering to only successful trajectories) can significantly degrade the model's performance.

We summarize our contributions as follows:

1. A quantitative analysis of the covariate shift experienced between student and expert models in the SWE agent domain.

2. *On-policy expert corrections*, a novel data generation technique that addresses the problem of covariate shift in multi-turn LM agent training.

3. A suite of experiments using the Qwen2.5-Instruct 7B and 32B models that achieve state-of-the-art performance within their classes on SWE-bench. We open-

source the resulting models and training data.

4. Ablations that highlight the need for further data filtering (beyond unit-test rejection sampling) in multi-turn LM agent training.

## 2. Related Works

**Training LM Agents.** LLM agent training paradigms can be broadly categorized into off-policy and on-policy methods. Off-policy training methods typically train via supervised fine-tuning on expert data, sometimes referred to as *imitation learning*. Within the category of imitation learning falls *distillation* (Hinton et al., 2015), a popular "teacher-student" paradigm where a smaller model replicates a larger one's behavior. On-policy approaches are typically based on reinforcement learning (RL). Several recent works have demonstrated the success of RL from verifiable rewards in LM agent training (Cao et al., 2025; Da et al., 2025), coding (Guo et al., 2024), and reasoning models (Gao et al., 2024). Similarly, rejection sampling fine-tuning (Zhang et al., 2023) improves agents by iteratively fine-tuning a model on successful, on-policy trajectories. Our method combines principles from both on-policy and off-policy methods, enjoying a mix of their benefits.

**Imitation Learning and Covariate Shift.** Imitation learning has a long history of literature spanning robotics (Abbeel & Ng, 2004), autonomous driving (Bojarski et al., 2016; Codevilla et al., 2018), and recently LLM agents (Chen et al., 2023; Anthropic, 2024; Yang et al., 2025). It is often used in domains where it is impractical to specify a reward function to train on, instead relying on expert demonstrations. A fundamental challenge in imitation learning is the problem of *covariate shift* (Ross & Bagnell, 2010), a problem afflicting classic methods such as behavioral cloning (Pomerleau, 1988) where inaccuracies resulting from distribution shifts compound in a multi-turn settings.

The dataset aggregation (DAgger) algorithm (Ross et al., 2011) is a seminal imitation learning method that addresses covariate shift. Our data generation techniques is inspired by DAgger by generating partially on-policy trajectories with some key adaptations to the multi-turn LLM agent setting. DAgger has spawned many followup works since its introduction. Most close to our approach are methods that "roll-in" with a student policy and then "roll-out" with an expert model to compute single-step cost-to-go values used for policy improvement (Ross & Bagnell, 2010; Chang et al., 2015). Our method instead uses the expert portion of the rollout directly for imitation learning since cost-to-go values are impractical to compute in the LM agent setting.

**LM Agents for SWE Tasks.** The integration of LLMs in software development has progressed from code generation to autonomous agentic behaviors that can solve complex

---

**Algorithm 1** OECs with random switching.

1: **Input:** student model $M^S$; expert model $M^E$; problem instance $\mathcal{P}$; switch index distribution $\mathcal{D}$.
2: switch $\sim \mathcal{D}$           *sample switch index*
3: $M \leftarrow M^S$           *start with student*
4: **for** step $= 1, \ldots, h$ **do**
5:      **if** step $=$ switch **then**
6:          $M \leftarrow M^E$         *switch to expert*
7:          history $\leftarrow$ **format**(history, $M^E$)   *rewrite in expert format*
8:      **end if**
9:      action $\leftarrow M$(history)      *sample current model*
10:      obs, state $\leftarrow T^{\mathcal{P}}$(state, action)    *transition environment*
11:      history $\leftarrow$ history $+$ [action, obs]    *append to history*
12: **end for**
13: **return** history

---

tasks. The SWE-bench benchmark (Jimenez et al., 2024) assesses an LLM's ability to resolve real-world software issues collected directly from GitHub. The task requires an agent to navigate a codebase, identify a problem described in an issue, and generate a code patch that addresses it. LM agents are typically deployed using a system, or scaffold, that creates an LM-friends interface that allows the model to easily read, write, and execute commands in the codebase. Agentless (Xia et al., 2024) was one of the early successful SWE scaffolds and used a procedural system for localizing and fixing bugs. Recently, scaffolds (e.g., SWE-agent (Yang et al., 2024) and Openhands (Wang et al., 2025)) have become more general, allowing LM agents to execute commands in arbitrary orders and leading to trajectories with many more turns, exacerbating the problem of covariate shift.

## 3. Methodology

We explore the setting of imitation learning in which training samples are generated via a stronger (potentially closed-source) model. We call the stronger model the *expert* and the model that we want to fine-tune as the *student*. In our setting, expert demonstrations take the form of a multi-turn trajectory $H = \{E_1, A_1, \ldots E_h, A_h\}$ alternating between environment $E_i$ and agent steps $A_i$. The agent steps include both thought tokens and the actions (i.e., tool calls) that affect the underlying environment. Each agent turn is generated by the LM agent by inputting the history $H_{1:i} = \{E_1, A_1, \ldots E_i\}$ of previous environment and agent turns.

### 3.1. On-policy Expert Corrections

The covariate shift experienced by an LM agent is not isolated to the state space of the underlying environment (e.g., the state of the SWE codebase). Since the entire history $h_i$ of a trajectory is used to generate a model's next action,

covariate shift can occur over the entire trajectory, including all previous environment turns, agent thought tokens, and actions. On-policy expert corrections (OECs) trajectories mitigate this by querying for expert demonstrations beginning part way through a student trajectory, at which point the history is necessarily on-policy. To achieve this, OECs are generated by rolling out trajectories using the student model and then, part way through a trajectory, swapping the underlying model to the expert. The swap between the student and expert model is performed randomly in order to cover as much of the on-policy state distribution as possible.

Algorithm 1 contains pseudocode for how OEC trajectories are generated. When the underlying model of the SWE-agent is switched from the student to the expert, we preserve the history of the trajectory, but rewrite the context of the model to match the scaffold formatting of the expert model. For example, system message and model-specific prompts are rewritten, and previous tool-calls are rewritten to match the expert's format. The expert model is not given any additional prompts or knowledge that previous steps were generated by different model. Section A contains an example OEC trajectory (condensed in Figure 1). The student model spends the first 29 turns of the rollout trying (but failing) to localize the bug (a common failure mode with open-source models). After switching to the expert, it finishes localization, writes and tests a patch, and successfully resolves the problem.

**Comparison to DAgger.** OECs enjoy several benefits over traditional DAgger. Firstly, they allow the use of rejection sampling based on environment feedback (in our case, unit tests), training signal that has been essential to the success of existing LM agent fine-tuning techniques. Traditional DAgger only rolls out the expert one extra step so it is not clear how one would apply verifiable rewards or otherwise assess the quality of the action. OECs also enjoy a much more efficient $O(T)$ dependency (where $T$ is the number of turns) to generate expert rollouts and train the student on those rollouts. In comparison, naive DAgger have an $O(T^2)$ dependency in the LM setting because it cannot directly reuse the KV cache from previous calls to the expert throughout a trajectory and thus must be recomputed for every turn. On the other hand, the KV cache in an OEC trajectory only needs to be recomputed a single-time, on the turn when switching from the student to the expert, and following turns can incrementally added to the existing KV cache. This computational improvement applies during expert generation and training. However, because of its differences, OEC trajectories do not inherit the same proven no-regret learning benefits of the traditional DAgger algorithm since the underlying state distribution is not completely on-policy.

### 3.2. Rejection Sampling and On-policy Masking

In order to fine-tune on a trajectory, each individual agent turn $A_i$ can be considered a training sample in the form $(H_{1:i}, A_i)$, i.e., what thinking and action the expert would perform given the previous turns. Given the autoregressive nature of LMs, we can speed up training by simultaneously training on all actions, masking out the loss of the environment steps of a trajectory (Ouyang & et al., 2022). After generating a set of OEC trajectories using Algorithm 1, we evaluate the resulting patches generated from the trajectories on each problem's associated unit tests. Following existing work (Pan et al., 2024; Yang et al., 2025), we filter out all trajectories that fail to pass all unit tests so that only successful trajectories are seen during fine-tuning. We also mask the on-policy (i.e., student) portion of each trajectory so that training is only performed on the expert portion of each trajectory, since it has been found that training on on-policy data can destabilize LM agent training (Pan et al., 2024). If an OEC trajectory is submitted before swapping from student to expert, we completely remove the sample from the fine-tuning set since the entire trajectory is on-policy and would be completely masked.

### 3.3. Filtering Repetitive Trajectories

Despite filtering down trajectories to only those that pass all unit-tests, we find that training on certain on-policy trajectories can severely degrade performance. As identified in previous works (Yang et al., 2025), open-source models (which we use as students) are often susceptible to falling into repetitive loops. We find that this most overwhelmingly presents itself as a model continuously reading one or more files in chunks over the course of many actions. Such trajectories have an outstanding impact on the gradient of the model since they often involve a sequence of nearly identical actions which are sometimes repeated until the trajectory times out, causing the model to overfit to and reinforce such negative behavior.

We find that even only partially on-policy trajectories, such as the case with on-policy expert correction (OEC) trajectories, are susceptible to such poor quality rollouts. We employ a simple programmatic filter for removing such trajectories by checking if either (1) an identical action (arguments included) is taken three times in a row (*identical actions*), or (2) any file-reading command is called 20+ times in a row (*repetitive file reading*).

## 4. Experiments

Our experiments are focused on answering the following questions.

1. How do on-policy expert corrections compare to other data generation techniques?

| Model | Agent Scaffold | Training size | Lite | Verified |
|---|---|---|---|---|
| OEC-SWE-32B* (ours) | SWE-Agent | 4,961 | 33% | 40.0% |
| SWE-smith-LM-32B* (Yang et al., 2025) | SWE-Agent | 5,016 | 28.3% | 36.4% |
| SWE-smith-LM-32B (Yang et al., 2025) | SWE-Agent | 5,016 | 30.7% | 40.2% |
| SWE-Gym-32B (Pan et al., 2024) | OpenHands | 491 | 15.3 % | 20.6% |
| Skywork-SWE-32B (Zeng et al., 2025) | OpenHands | 8,447 | — | 38.0% |
| OEC-SWE-7B* (ours) | SWE-Agent | 8,943 | 17% | 20.8% |
| SWE-smith-LM-7B (Yang et al., 2025) | SWE-Agent | 5,016 | 11.7% | 15.2% |
| SWE-Gym-7B (Pan et al., 2024) | OpenHands | 491 | 10% | 10.6% |
| SkyRL-Agent-7B-v0 (Cao et al., 2025) | OpenHands | 3,680+ | — | 14.6% |

*Table 1.* Resolution rates on SWE-bench for various models fine-tuned on top of the Qwen2.5-Coder family. Model names marked with * are based on our own evaluations; others are taken as reported in existing works. *# Samples* for `SkyRL-Agent-7B-v0` is marked with a + because it is fine-tuned on top of `OpenHands-7B-Agent` which is already fine-tuned on an unknown number of additional SWE agent trajectories.

2. How important is on-policy masking and repetition filtering to the success of on-policy expert corrections?

3. How much covariate shift is experienced between a student and expert agents?

4. What sorts of qualitative behavior changes in SWE agents do different methods lead to?

### 4.1. Experiment Setup

**Student and expert models.** We explore the impact of different data generation techniques in a 7B and 32B parameter setting. We choose student models that are already relatively performant on SWE-agent tasks so that the on-policy portions of trajectories are high quality. In the 7B setting, our student model, which we call `Student-7B`, is trained using the recipe from (Yang et al., 2025) by taking `Qwen2.5-Coder-7B-Instruct` and fine-tuning on the 5,000 open-source `Claude-3.7-Sonnet` (Anthropic, 2025) expert SWE-agent trajectories released in (Yang et al., 2025). For cheaper experimentation, we use the open-source `SWE-agent-LM-32B` (Yang et al., 2025) model as our expert in this setting.

Our 32B student, `Student-32B`, is trained in a similar way, starting with `Qwen2.5-Coder-32B-Instruct` and training on 2,000 of the expert `Claude-3.7-Sonnet` trajectories so that 3,000 trajectories remain held out for later experiments. In this setting, we use `Claude-3.7-Sonnet` as our expert so that our trajectories are comparable to the 3,000 held out trajectories (which also use `Claude-3.7-Sonnet`).

**Training problem instances.** We use SWE-smith (Yang et al., 2025) problem instances to generate training trajectories of all types. The primary subset of SWE-smith instances that we train on is obtained by extracting the problem instances from the 5,000 open-source `Claude-3.7-Sonnet` expert trajectories, since we know these instances can be feasibly solved by language models. This process results in 4,121 unique problem instances. In

the 7B setting, we use all of these problem instances to generate training trajectories. In the 32B setting, we filter this down to the 2,308 instances that `Student-32b` has not already been trained on since we found (as reported in (Pan et al., 2024)) that training on too many trajectories from a single instance hurt performance.

**Agent scaffold.** We use the SWE-agent scaffold to generate training trajectories of all types and adopt the system prompts from (Yang et al., 2025) for all of our models. We pick SWE-Agent for it's popularity, multi-turn structure, and performance. Since the scaffold can vary (in system prompt, problem statement, and tool calls) between model types, SWE-agent trajectories are modified to match the format of the student model during training. Additionally, when on-policy expert corrections switch from a student to expert partway through a rollouts, the agent's history is modified to match the expert's format. This includes patching the system prompt and problem statement as well as reformatting tool calls.

**Methods.** In our experiments, we generate OECs by randomly switching from student to expert sampled from the uniform distribution $U(0, 30)$. We compare on-policy expert corrections (OECs) against two different trajectory generation techniques. The first of these is *behavioral cloning (BC)* (Pomerleau, 1988), where trajectories are rolled out by the expert model (`SWE-agent-LM-32B` or `Claude-3.7-Sonnet` in our case) from the very beginning of the trajectory, a popular method employed in many prior works on SWE-agent training (Yang et al., 2025; Pan et al., 2024). The other method, which we call *on-policy* trajectory generation, is where trajectories are entirely rolled out by the student model, similar to how trajectories are rolled out on-policy during RL. All of the methods are combined with rejection sampling fine-tuning and repetition filtering and we test scaling with respect to dataset size by fine-tuning the student models from scratch on varying numbers of trajectories of each type.

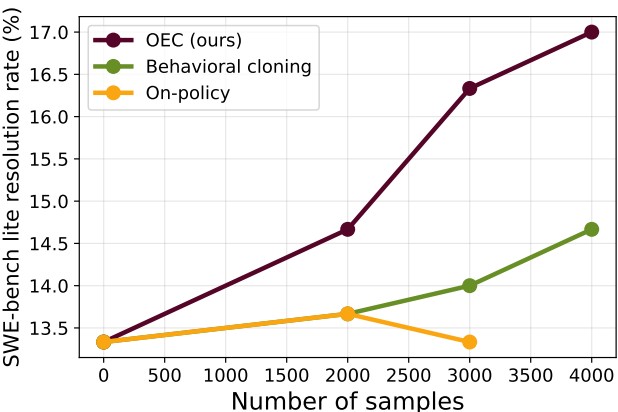

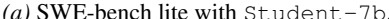

*(a)* SWE-bench lite with `Student-7b`.

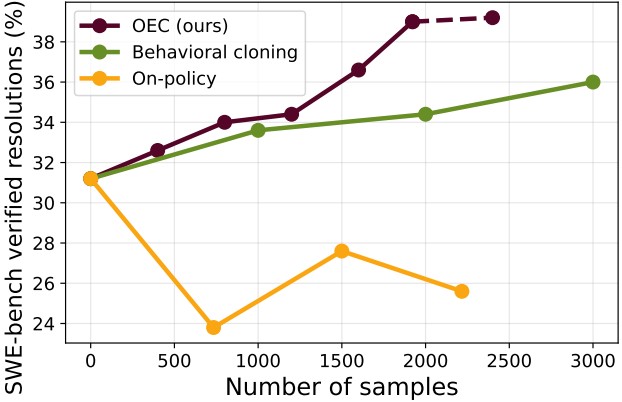

*(b)* SWE-bench verified with `Student-32B`.

*Figure 2.* Resolution rates on SWE-bench while scaling dataset sizes between three trajectory collection techniques: (1) OECs (ours), (2) behavioral cloning, and (3) on-policy trajectories. Each point is obtained by fine-tuning the corresponding student model from scratch on the number of trajectories of that type. All techniques are combined with repetition filtering and rejection sampling.

| Method | Mean | Median |
|---|---|---|
| BC (expert) | 22,812 | 18,585 |
| OEC (expert) | 21,116 | 18,014 |
| OEC (student) | 5,444 | 4,372 |

*Table 2.* Mean and median token counts for expert and student-generated portions of trajectories in the 32B scaling experiments in Figure 2b.

## 4.2. Results

Figure 1 shows the performance of our models against existing models finetuned with BC or RL on top of the Qwen2.5-Coder family, showing that we achieve SOTA performance within model classes in the 7B and 32B setting. Note that we reran evaluations for some existing models on our setup and found lower performance than reported, in which case we present both values.

**Comparison between trajectory generation techniques.** Figure 2a shows dataset size scaling results for OEC, BC, and on-policy trajectory generation methods in the 7B setting. In this setting, our expert is `SWE-Smith-LM-32B` and we gather each trajectory type by doing three passes on the 4,121 SWE-smith instances. Our initial experiments found that it was important to regularize the training of `Student-7B` with some of the orig-

inal `Claude-3.7-Sonnet` trajectories used to train `Student-7B`. Thus, each model in Figure 2a has an additional 1,000 of the original `Claude-3.7-Sonnet` expert trajectories mixed in during fine-tuning. The results demonstrate that training with OECs leads to larger performance gains than BC while training with on-policy trajectories leads to little (if any) improvement.

We similarly found that in the 32B setting, a combination of OEC and BC samples proved most effective. Thus, in these experiments the OEC trajectories are combined with an equal amount of BC trajectories and the x-axis represents the total number of samples (OEC + BC) used for fine-tuning. The OEC trajectories for all but the final data point in Figure 2b were collected through a single pass on the 2,308 SWE-smith instances with `Student-32B` as student and `Claude-3.7-Sonnet` as expert resulting in a 62.3% resolution rate. We sampled an additional 2,476 SWE-Smith problem instances to generate more OEC trajectories, but these instances proved to be significantly more challenging at a 12% resolution rate. These additional OEC trajectories are only included for the final point in the *OEC* line and since this additional set of problem instances comes from a different problem distribution (which may bias the results) than the *BC* and *On-policy* lines, we notate the final point connected with a dashed line. The BC samples are sourced from the remaining 3,000 `Claude-3.7-Sonnet` expert trajectories and the on-policy trajectories are generated through three passes on SWE-smith instances at a 36% resolution rate.

Figure 2b shows dataset size scaling results for the 32B setting demonstrating similar findings as in the 7B setting. OEC trajectories provide a substantial boost over pure BC trajectories with a mix of 961 OEC and 961 BC trajectories resulting in resolving an additional 23 (4.6%) SWE-bench verified instances in comparison to fine-tuning on purely 2000 BC trajectories. Despite the use of unit-test rejection sampling and repetition filtering, fine-tuning on fully on-policy trajectories severely decreased the performance of the model in the 32B setting, similar to observations made in prior work (Pan et al., 2024).

To better understand the compute cost of different trajectory generation methods, we analyzed the distribution of expert and student tokens used in the 32B scaling experiments

from Figure 2b. As shown in **??**, BC trajectories require an average of 22,812 expert tokens per trajectory with a median of 18,585. In contrast, OEC trajectories require an average of 21,116 expert tokens with a median of 18,014, while additionally using an average of 5,444 student tokens with a median of 4,372. Thus, OEC reduces expert token usage by 1,696 tokens per trajectory on average, corresponding to an approximately 8% reduction relative to BC. This reduction arises because OEC trajectories are partially generated by the student before the expert provides a correction. Although total trajectory length is somewhat larger for OEC once student tokens are included, the expensive expert-token budget is reduced, suggesting that OEC can provide a modest efficiency benefit over BC in addition to its performance gains.

We also ran an experiment to compare OEC against Agent-RLVR (Da et al., 2025), a recent RL-based approach for fine-tuning LM agents in the SWE agent domain. We evaluate Agent-RLVR under the same conditions as the 7B scaling experiments in Figure 2a: starting from the same 7B student model, using rollouts from the same problem distribution, and fine-tuning on 2,000 training samples. On SWE-bench Lite, OEC achieves a 14.7% resolution rate, while Agent-RLVR achieves 13.7%, matching the performance of the BC and on-policy baselines at this data scale. These results suggest that, in addition to outperforming BC and direct on-policy fine-tuning in our main scaling experiments, OEC achieves comparable or slightly better gains than a recent RL-based method in this setting.

**A combination of OEC and BC trajectories performs best.** Our early experiments indicated that a mixture of OEC trajectories and BC trajectories for fine-tuning outperformed either one alone. Figure 3 shows scaling results for various mixtures of OEC and BC samples. The blue line varies the number of BC samples while keeping the number of OEC samples fixed at 961 while the red line varies the number of OEC samples while keeping the number of BC samples fixed at 961. The two lines coincide at $x = 1,922$ where samples are evenly split between OEC and BC trajectories. The results demonstrate the importance of both OEC samples and BC samples for best performance. The final point in this plot comes from a mixture of 961 OEC trajectories and 2000 BC trajectories and represents the highest performing model across all of our experiments.

**Filtering repetitive trajectories and on-policy masking is critical.** In this section we study the importance of filtering out repetitive training trajectories and performing on-policy masking during fine-tuning with OEC trajectories. Table 3 shows the resulting performance impact on the model without filtering, representing an absolute 6% resolution rate difference in the 7B setting and a 3.6% difference in the 32B setting. Notably, we found the problem of repetitive

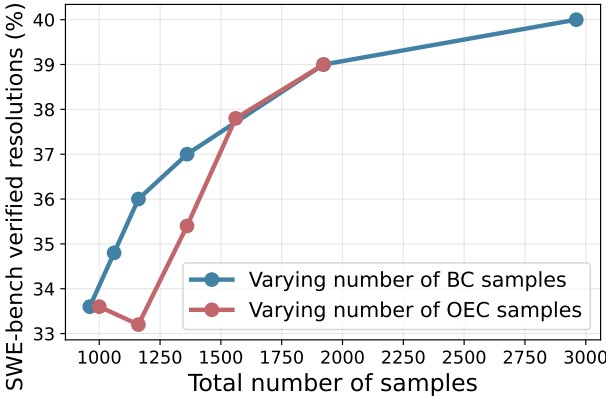

*Figure 3.* Varying the mixture between BC and OEC trajectories.

| Student model | No training | OEC | No mask | No filter |
|---|---|---|---|---|
| Student-7b | 13.3% | 17% | 17% | 11% |
| Student-32b | 31.2% | 40% | 31.8% | 36.4% |

*Table 3.* Ablations for removing on-policy masking (*No mask*) and repetition filtering (*No filter*) while fine-tuning on on-policy corrections (OECs). `Student-32b` is evaluated on SWE-bench verified and `Student-7b` on SWE-bench lite.

trajectories more prevalent in the 7B setting in which both the student and expert model are smaller models, which are known to be susceptible repetitive loops (Yang et al., 2025). In the 7B setting, 8.46% (167 from *repetitive file reading* and 93 from *identical actions*) samples needed to be filtered and in the 32B setting, 3.63% (43 from *repetitive file reading* and 2 from *identical actions*). Notably, repetition filtering is performed on top of rejection sampling based on unit tests, highlighting the importance of repetition filtering *in addition to* rejection sampling. Some of the training problems are easy enough that even when agents get stuck in repetitive loops, they still sometimes succeed in solving the problem and would ordinarily find themselves in the positive training group (a problem that would afflict other on-policy approaches like reinforcement learning).

Table 3 also shows that on-policy masking makes no difference in the 7B setting but is crucial in the 32B setting, essentially negating any benefit from the expert portion of the data. This matches our results on fine-tuning with purely on-policy trajectories: having little to no affect in the 7B setting, but severely decreasing performance in the 32B setting.

**Later switching improves the model more.** In order to study the impact of *when* OEC trajectories are switched from student to expert, we partition our full set of 1,200 OEC trajectories into three subsets based on switch index. We chose ranges in such a way to create as even of a partition of samples as possible, resulting in ranges 0-6, 7-15, and 16-30 (the reason there are more samples with smaller switch

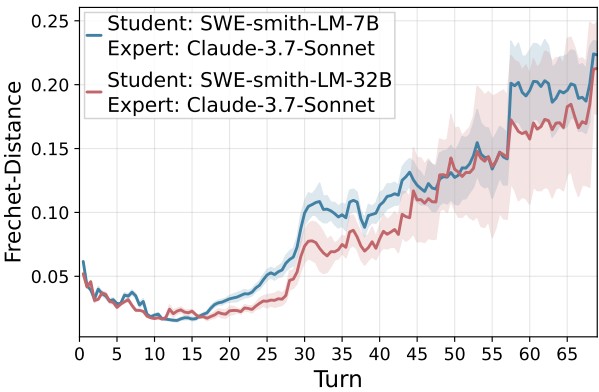

*Figure 4.* Covariate shift between student and expert throughout a set of trajectories.

indices is that a larger switch index increases the likelihood that a trajectory terminate before switching to the expert, in which case we discard it). We used the minimum number of samples of 372 across all partitions to create three disjoint training sets. We then randomly sampled 372 BC samples and added them to each set to get a total of 744 training samples in each set. The SWE-bench verified resolution rate after fine-tuning on these the datasets for intervals 0-6, 7-15, and 16-30, respectively, are 31%, 32.4%, and 32.6%.

### 4.3. Covariate Shift between the student and expert

In this section, we empirically study the amount of covariate shift experienced between student and expert models, finding that a the covariate shift between a student and expert grows throughout a trajectory, even if it has been finetuned through imitation learning. For students, we use `SWE-smith-LM-7B` and `SWE-smith-LM-32B` that are both trained through behavioral cloning on the expert `Claude-3.7-Sonnet` trajectories. We select 10 problem instances from SWE-Smith at random, and generate 50 rollouts for each model with temperature 0.1 to get a distribution over trajectories.

Conceptually, the *state* of a SWE-agent rollout includes not just the state of the underlying codebase, but the entire history of the rollout since that is what is used as input to the model to generate the next action. In order to approximate the covariate shift in the history along a trajectory, we first embed the history at each timepoint into a representation space. We use the Qwen3-8B embedding model (Zhang et al., 2025), transforming each trajectory into a continuous-space representation of dimension $n \times 4096$, where $n$ is the number of turns in the rollout.

For each turn we construct replicates of 10 rollout embeddings each, bootstrapping if necessary; within each replicate, we fit a Gaussian over the embeddings. To quantify

covariate shift between student and expert, we compute the multivariate Gaussian Fréchet Distance (FD) between all pairs of student-expert replicates, averaging across pairs to obtain a turnwise divergence score. This procedure follows a similar paradigm to Fréchet Inception Distance (FID) (Heusel et al., 2017), where FD over embedding distributions captures representational drift, while also accounting for the multi-turn nature of agent interactions. To provide a baseline reference for the FD within a set of trajectories we also computed the FD between randomly sampled subsets of the expert trajectories which we found to remain very low (see Figure 6).

Figure 4 shows the FD (see Figure 5 for KL-Divergence, showing similar trends) between both students and expert with shading indicating 95% confidence intervals. The FD briefly dips until around turn 10, sharply increases until turn 30, and then steadily continues to rise. These results empirically demonstrate a sharp difference between student and expert trajectories with worsening covariate shift throughout a trajectory, even though the student has been trained to imitate the expert through behavioral cloning.

### 4.4. Qualitative Analysis

Using GPT-5, we conduct an LLM-as-a-judge analysis to examine the failure reasons across different models. Our procedure follows Yang et al. (2024), which reported 87% alignment of automated judgments with human labeling over a given set of failure categories. To start, we define buckets capturing recurring failure patterns in software engineering tasks, informed by heuristics and randomly sampled set of trajectories. The first group of buckets captures failures in trajectories that were submitted: this includes providing a wrong solution, making syntax errrors that prevented execution, modification of the incorrect file, non-instruction following, missing of a model edge case, or another reason. Likewise, the second group of buckets captures non-submission failures, including errors in tool-calling, hitting context horizons, and getting stuck in loops.

For each of the models Table 4, we filter to only unresolved instances of the SWE-smith instances and collect the last 20 turns of each rollout. Then, we guide the judge (prompt give in Section C) with a system prompt providing descriptions of failure buckets and SWE-Agent scaffold format, and feed in each trunated trajectory input. The judge produces a paragraph reasoning, and then an ultimate classification of one failure mode per instance.

## 5. Conclusion

We introduce a novel, partially on-policy data generation technique, called *on-policy expert corrections (OECs)* to address the problem of covariate shift in imitation learning for

| Model / Dataset | Resolve (%) | Submitted? | | Errors for Submitted Patches | | | | | | Errors for No Submission | | |
|---|---|---|---|---|---|---|---|---|---|---|---|---|
| | | Yes | No | Wrong Solution | Syntax Error | Wrong File | Instruct Follow | Edge Case | Other | Tool Use | Long-Context | Stuck in Loop |
| STUDENT-32B | 31.2% | 78.7% | 21.3% | 44.7% | 7.1% | 2.7% | 40.8% | 0.8% | 3.9% | 24.6% | 17.4% | 58.0% |
| + 3000 BC | 36.0% | 82.4% | 17.6% | 59.7% | 6.5% | 4.9% | 22.8% | 1.1% | 4.9% | 33.9% | 16.1% | 50.0% |
| + 961 OEC + 961 BC | 39.0% | 82.2% | 17.8% | 50.4% | 6.4% | 4.0% | 32.0% | 1.6% | 5.6% | 25.9% | 22.2% | 51.9% |
| + 961 OEC + 2000 BC | 40.0% | 81.9% | 18.1% | 59.8% | 5.7% | 2.9% | 24.2% | 1.6% | 5.7% | 27.8% | 18.5% | 53.7% |
| ... NO REPETITION FILTERING | 36.4% | 86.4% | 13.6% | 58.0% | 4.0% | 5.8% | 27.4% | 1.8% | 2.9% | 37.2% | 11.6% | 51.2% |
| ... NO ON-POLICY MASKING | 31.8% | 82.5% | 17.5% | 39.1% | 4.3% | 3.9% | 47.0% | 1.4% | 4.3% | 30.5% | 16.9% | 52.5% |

*Table 4.* LLM-as-a-judge qualitative analysis of unresolved (i.e., failed) SWE-bench verified trajectories for different models fine-tuned throughout our experiments. "Submitted" describes if the model was able to sucessfully generate a patch for the problem. Resolve rates are on SWE-Bench verified.

multi-turn LM agents. Our technique combines the strengths of several existing paradigms for LM agent training: the relevance of on-policy training from RL, expert data from imitation learning methods, and rejection sampling from training with verifiable rewards. Our experiments highlight the limitations of relying on either purely on-policy training or purely expert demonstrations. Moreover, our experiments highlight the importance of evaluating data quality beyond verifiable rewards, showing that a small proportion of low-quality, positive trajectories can greatly destabilize learning. Our experiments focus on the SWE agent setting, but it will be important for future work to test our findings in other multi-turn LM agent domains, especially as LM agents are used for more complex and long-horizon tasks (Kwa et al., 2025) and the problem of covariate shift becomes increasingly severe.

**Limitations.** Although they have their clear benefits, OECs are not known to benefit from the same no-regret learning guarantees as traditional DAgger. Also, as fine-tuning is performed on a set of OEC trajectories, the on-policy portions of the trajectories become increasingly off-policy, potentially limiting their benefit. This could be mitigated by doing multiple intermediate rounds of OEC generation or by generating new OEC trajectories in an online fashion. Like other imitation learning approaches, OEC trajectories require a source of the expert trajectories (unlike other approaches such RL). In this work, we explored the setting in which a stronger model provides the expert trajectories, whoever, our ideas could be extended to human expert data, using increased test-time compute (e.g., best-of-N (Brown et al., 2020) or tree-of-thought (Yao et al., 2023)), or privileged information (e.g., hints (Nath et al., 2025)) to generate OEC trajectories.

## Impact Statement

This paper presents work whose goal is to improve the performance and robustness of LM agents in long-horizon settings. Since LM agents are being increasingly deployed in different domains, there are many potential societal consequences of our work, none of which we feel are specific to the methods or evaluations proposed in our work.

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

# A. An Example On-policy Expert Correction Trajectory

Here we include the full trajectory transcript for a real example of an on-policy expert correction trajectory found in our training data. The student model, `Student-32B` (red boxes), spends the first 29 turns of the rollout trying (but failing) to localize the bug (a common failure mode with open-source models). After switching to the expert, `Claude-3.7-Sonnet` (green boxes), it finishes localization, writes and tests a patch, and successfully resolves the instance.

---

**Problem Statement**

**# YamlLint ignores list configuration doesn't work properly**
I'm having an issue with the `ignore` configuration in YamlLint. When I try to use a
list of patterns to ignore files, it doesn't work as expected.

**## Reproduction**
have a configuration file with an ignore list like this:
```yaml
extends: default
ignore:
- "*.dont-lint-me.yaml"
  - "/bin/"
  - "!/bin/*.lint-me-anyway.yaml"
```
But when I run yamllint, it still lints all files, including those that should be
ignored.

**## Expected behavior**
Files matching the patterns in the ignore list should be excluded from linting.

**## Actual behavior**
All files are being linted regardless of the ignore patterns.
I've also tried using the ignore configuration in a rule-specific context:
```yaml
rules:
  key-duplicates:
    ignore:
      - "/ign-dup"
  trailing-spaces:
    ignore:
      - "ign-trail"
      - "!*.lint-me-anyway.yaml"
```
But this also doesn't work when the ignore is specified as a list.
It seems like the ignore functionality only works when using a string, not a list of
patterns. This is inconsistent with the documentation which suggests that a list of
patterns should work.

---

**Student Model Agent**

I'll help you implement the necessary changes to fix the YamlLint ignore list configuration issue. Let's follow the steps you outlined.
## Step 1: Find and read code relevant to the PR description
First, let's explore the repository structure to un...

```
find /testbed -type f -name "*.py" | grep -v "__pycache__" | sort
```

---

**Environment**

```
/testbed/docs/conf.py
/testbed/setup.py
/testbed/tests/__init__.py
/testbed/tests/common.py
/testbed/tests/rules/__init__.py
/testbed/tests/rules/test_anchors.py
/testbed/tests/rules/test_braces.py
/testbed/tests/rules/test_brackets.py
...
```

---

**Student Model Agent**

Now, let's look for files that might be related to the ignore functionality. Based on the PR description, we need to focus on how the ignore patterns are processed.

```
grep -r "ignore" --include="*.py" /testbed/yamllint/
```

```
/testbed/yamllint/config.py:        self.ignore = None
/testbed/yamllint/config.py:    def is_file_ignored(self, filepath):
/testbed/yamllint/config.py:        return self.ignore and self.ignore.match_file(filepath)
/testbed/yamllint/co...
```

**Student Model Agent**

Let's also check for any imports of the 'pathspec' module, which seems to be used for pattern matching:

```
grep -r "import pathspec" --include="*.py" /testbed/
```

**Environment**

```
/testbed/yamllint/config.py:import pathspec
```

**Student Model Agent**

Now, let's look at the specific parts of the code that handle the ignore configuration. Let's examine the 'config.py' file, which seems to be the main file handling the ignore patterns:

```
str_replace_editor view /testbed/yamllint/config.py
```

**Environment**

```
Here's the result of running `cat -n` on /testbed/yamllint/config.py:
     1          # Copyright (C) 2016 Adrien Vergé
     2          #
     3          # This program is free software: you can redistribute it and/or modify
     4          # it under the terms of the GNU...
```

**Student Model Agent**

Now, let's look at the linter.py file to understand how the ignore functionality is used:

```
str_replace_editor view /testbed/yamllint/linter.py
```

**Environment**

```
Here's the result of running `cat -n` on /testbed/yamllint/linter.py:
     1          # Copyright (C) 2016 Adrien Vergé
     2          #
     3          # This program is free software: you can redistribute it and/or modify
     4          # it under the terms of the GNU...
```

**Student Model Agent**

Let's also check the CLI file to understand how the linter is invoked:

```
str_replace_editor view /testbed/yamllint/cli.py  --view_range 1 50
```

**Environment**

```
Here's the result of running `cat -n` on /testbed/yamllint/cli.py:
     1          # Copyright (C) 2016 Adrien Vergé
     2          #
     3          # This program is free software: you can redistribute it and/or modify
     4          # it under the terms of the GNU Ge...
```

**Student Model Agent**

Let's look at the rest of the CLI file:

```
str_replace_editor view /testbed/yamllint/cli.py  --view_range 50 100
```

**Environment**

```
Here's the result of running `cat -n` on /testbed/yamllint/cli.py:
    50              @staticmethod
    51              def parsable(problem, filename):
    52                  return (f'\{filename\}:\{problem.line\}:\{problem.column\}: '
    53                          f'[\{pr...
```

**Student Model Agent**

```
str_replace_editor view /testbed/yamllint/cli.py  --view_range 100 150
```

**Environment**

```
Here's the result of running `cat -n` on /testbed/yamllint/cli.py:
    100                 args_format = 'colored'
    101
    102             for problem in problems:
    103                 max_level = max(max_level, PROBLEM_LEVELS[problem.level])
    104          ...
```

**Student Model Agent**

```
str_replace_editor view /testbed/yamllint/cli.py  --view_range 150 200
```

**Environment**

```
Here's the result of running `cat -n` on /testbed/yamllint/cli.py:
    150             files_group.add_argument('files', metavar='FILE_OR_DIR', nargs='*',
    151                                     default=(),
    152                                     help='...
```

**Student Model Agent**

```
str_replace_editor view /testbed/yamllint/cli.py  --view_range 200 250
```

**Environment**

```
Here's the result of running `cat -n` on /testbed/yamllint/cli.py:
    200                 elif os.path.isfile(user_global_config):
    201                     conf = YamlLintConfig(file=user_global_config)
    202                 else:
    203                     conf = ...
```

**Student Model Agent**

Now, let's look at the 'linter.py' file to understand how the ignore functionality is used:

```
str_replace_editor view /testbed/yamllint/linter.py  --view_range 1 50
```

**Environment**

```
Here's the result of running `cat -n` on /testbed/yamllint/linter.py:
    1           # Copyright (C) 2016 Adrien Vergé
    2           #
    3           # This program is free software: you can redistribute it and/or modify
    4           # it under the terms of the GNU...
```

**Student Model Agent**

```
str_replace_editor view /testbed/yamllint/linter.py  --view_range 50 100
```

**Environment**

```
Here's the result of running `cat -n` on /testbed/yamllint/linter.py:
    50          def message(self):
    51              if self.rule is not None:
    52                  return f'\{self.desc\} (\{self.rule\})'
    53              return self.desc
    54
...
```

**Student Model Agent**

```
str_replace_editor view /testbed/yamllint/linter.py  --view_range 100 150
```

**Environment**

```
Here's the result of running `cat -n` on /testbed/yamllint/linter.py:
   100                    rules = [item[5:] for item in items][1:]
   101                    if len(rules) == 0:
   102                        self.rules.clear()
   103            ...
```

**Student Model Agent**

```
str_replace_editor view /testbed/yamllint/linter.py  --view_range 150 200
```

**Environment**

```
Here's the result of running `cat -n` on /testbed/yamllint/linter.py:
   150
   151                disabled.process_comment(elem)
   152                if elem.is_inline():
   153                    disabled_for_line.process_comment(elem)
   15...
```

**Student Model Agent**

```
str_replace_editor view /testbed/yamllint/linter.py  --view_range 200 250
```

**Environment**

```
Invalid `view_range`: [200, 250]. Its second element `250` should be smaller than the number of lines in the
↪  file: `236`
```

**Student Model Agent**

```
str_replace_editor view /testbed/yamllint/linter.py  --view_range 200 236
```

**Environment**

```
Here's the result of running `cat -n` on /testbed/yamllint/linter.py:
   200                # Insert the syntax error (if any) at the right place...
   201                if (syntax_error and syntax_error.line <= problem.line and
   202                    ...
```

**Student Model Agent**

Now, let's look at the 'config.py' file to understand how the ignore patterns are processed:

```
str_replace_editor view /testbed/yamllint/config.py  --view_range 1 50
```

**Environment**

```
Here's the result of running `cat -n` on /testbed/yamllint/config.py:
   1      # Copyright (C) 2016 Adrien Vergé
   2      #
   3      # This program is free software: you can redistribute it and/or modify
   4      # it under the terms of the GNU...
```

**Student Model Agent**

```
str_replace_editor view /testbed/yamllint/config.py  --view_range 50 100
```

**Environment**

```
Here's the result of running `cat -n` on /testbed/yamllint/config.py:
   50          def is_yaml_file(self, filepath):
   51              return self.yaml_files.match_file(os.path.basename(filepath))
   52
   53          def enabled_rules(self, f...
```

**Student Model Agent**

```
str_replace_editor view /testbed/yamllint/config.py  --view_range 100 150
```

**Environment**

```
Here's the result of running `cat -n` on /testbed/yamllint/config.py:
   100              if 'ignore' in conf and 'ignore-from-file' in conf:
   101                  raise YamlLintConfigError(
   102                      'invalid config: ignore and ig...
```

**Student Model Agent**

```
str_replace_editor view /testbed/yamllint/config.py  --view_range 150 200
```

**Environment**

```
Here's the result of running `cat -n` on /testbed/yamllint/config.py:
   150              if isinstance(conf, dict):
   151                  if ('ignore-from-file' in conf and not isinstance(
   152                          conf['ignore-from-file'], pathspec.path...
```

**Student Model Agent**

```
str_replace_editor view /testbed/yamllint/config.py  --view_range 200 250
```

**Environment**

```
Invalid `view_range`: [200, 250]. Its second element `250` should be smaller than the number of lines in the
↪  file: `238`
```

**Student Model Agent**

```
str_replace_editor view /testbed/yamllint/config.py  --view_range 200 238
```

**Environment**

```
Here's the result of running `cat -n` on /testbed/yamllint/config.py:
   200                          type(flag) not in options[optkey]
   201                          for flag in conf[optkey])):
   202                      raise Y...
```

**Student Model Agent**

Now, let's look at the 'linter.py' file to understand how the ignore functionality is used:

```
str_replace_editor view /testbed/yamllint/linter.py  --view_range 1 50
```

**Environment**

```
Here's the result of running `cat -n` on /testbed/yamllint/linter.py:
     1       # Copyright (C) 2016 Adrien Vergé
     2       #
     3       # This program is free software: you can redistribute it and/or modify
     4       # it under the terms of the GNU...
```

**Student Model Agent**

```
str_replace_editor view /testbed/yamllint/linter.py  --view_range 50 100
```

**Environment**

```
Here's the result of running `cat -n` on /testbed/yamllint/linter.py:
    50          def message(self):
    51              if self.rule is not None:
    52                  return f'\{self.desc\} (\{self.rule\})'
    53              return self.desc
    54
...
```

**Student Model Agent**

```
str_replace_editor view /testbed/yamllint/linter.py  --view_range 100 150
```

**Environment**

```
Here's the result of running `cat -n` on /testbed/yamllint/linter.py:
   100                       rules = [item[5:] for item in items][1:]
   101                       if len(rules) == 0:
   102                           self.rules.clear()
   103               ...
```

**Student Model Agent**

```
str_replace_editor view /testbed/yamllint/linter.py  --view_range 150 200
```

**Environment**

```
Here's the result of running `cat -n` on /testbed/yamllint/linter.py:
   150
   151                   disabled.process_comment(elem)
   152               if elem.is_inline():
   153                   disabled_for_line.process_comment(elem)
   15...
```

**Student Model Agent**

```
str_replace_editor view /testbed/yamllint/linter.py  --view_range 200 236
```

**Environment**

```
Here's the result of running `cat -n` on /testbed/yamllint/linter.py:
   200               # Insert the syntax error (if any) at the right place...
   201               if (syntax_error and syntax_error.line <= problem.line and
   202                     ...
```

**Student Model Agent**

Now, let's look at the 'config.py' file to understand how the ignore patterns are processed:

```
str_replace_editor view /testbed/yamllint/config.py  --view_range 1 50
```

**Environment**

```
Here's the result of running `cat -n` on /testbed/yamllint/config.py:
     1       # Copyright (C) 2016 Adrien Vergé
     2       #
     3       # This program is free software: you can redistribute it and/or modify
     4       # it under the terms of the GNU...
```

**Student Model Agent**

```
str_replace_editor view /testbed/yamllint/config.py  --view_range 50 100
```

**Environment**

```
Here's the result of running `cat -n` on /testbed/yamllint/config.py:
    50           def is_yaml_file(self, filepath):
    51               return self.yaml_files.match_file(os.path.basename(filepath))
    52
    53           def enabled_rules(self, f...
```

**Student Model Agent**

```
str_replace_editor view /testbed/yamllint/config.py  --view_range 100 150
```

**Environment**

```
Here's the result of running `cat -n` on /testbed/yamllint/config.py:
   100                if 'ignore' in conf and 'ignore-from-file' in conf:
   101                    raise YamlLintConfigError(
   102                        'invalid config: ignore and ig...
```

**Expert Model Agent**

```
str_replace_editor view /testbed/yamllint/config.py  --view_range 150 200
```

**Environment**

```
Here's the result of running `cat -n` on /testbed/yamllint/config.py:
   150            if isinstance(conf, dict):
   151                if ('ignore-from-file' in conf and not isinstance(
   152                        conf['ignore-from-file'], pathspec.path...
```

**Expert Model Agent**

```
str_replace_editor view /testbed/yamllint/config.py  --view_range 200 238
```

**Environment**

```
Here's the result of running `cat -n` on /testbed/yamllint/config.py:
   200                                type(flag) not in options[optkey]
   201                                for flag in conf[optkey])):
   202                            raise Y...
```

**Expert Model Agent**

Based on the code review, I've identified the issue. In the 'config.py' file, there are two places where the 'ignore' configuration is processed:
1. In the 'parse' method (lines 114-119), it only accepts a string for the top-level 'ignore' configura...

```
str_replace_editor create /testbed/reproduce_issue.py --file_text '#!/usr/bin/env python3

import os
import tempfile
import sys

# Add the current directory to the Python path
sys.path.insert(0, os.path.abspath(os.path.dirname(__file__)))

from yamll...
```

**Environment**

```
File created successfully at: /testbed/reproduce_issue.py
```

**Expert Model Agent**

Let's run the script to confirm the issue:

```
python /testbed/reproduce_issue.py
```

**Environment**

```
Error: invalid config: ignore should contain file patterns

Error with rule-specific config: invalid config: ignore should contain file patterns
```

**Expert Model Agent**

Great, we've confirmed the issue. The error message "invalid config: ignore should contain file patterns" is being raised because the code expects a string but we're providing a list.
Now, let's implement the fix. We need to modify the 'config.py' f...

```
str_replace_editor str_replace /testbed/yamllint/config.py  --old_str '        elif '"'"'ignore'"'"' in
↪   conf:
            if not isinstance(conf['"'"'ignore'"'"'], str):
                raise YamlLintConfigError(
                    '"'"'invalid con...
```

---

**Environment**

```
The file /testbed/yamllint/config.py has been edited. Here's the result of running `cat -n` on a snippet of
↪  /testbed/yamllint/config.py:
   109                         raise YamlLintConfigError(
   110                             'invalid...
```

---

**Expert Model Agent**

Now, let's also fix the rule-specific 'ignore' configuration in the 'validate_rule_conf' function:

```
str_replace_editor str_replace /testbed/yamllint/config.py   --old_str '        elif ('"'"'ignore'"'"' in
↪  conf and
             not isinstance(conf['"'"'ignore'"'"'], pathspec.pathspec.PathSpec)):
         if not isinstance(conf['"'"'ignore'"'"'...
```

---

**Environment**

```
The file /testbed/yamllint/config.py has been edited. Here's the result of running `cat -n` on a snippet of
↪  /testbed/yamllint/config.py:
   163                        'invalid config: ignore-from-file should contain '
   16...
```

---

**Expert Model Agent**

Now, let's run the script again to see if our changes fixed the issue:

```
python /testbed/reproduce_issue.py
```

---

**Environment**

```
Config created successfully!

Testing file ignoring:
test.yaml ignored: False
test.dont-lint-me.yaml ignored: True
bin/test.yaml ignored: False
bin/test.lint-me-anyway.yaml ignored: False

Rule-specific config created successfully!
```

---

**Expert Model Agent**

Great! The script now runs without errors, which means our changes fixed the issue. However, there seems to be an issue with the file ignoring logic. The 'bin/test.yaml' file should be ignored according to the pattern '/bin/', but it's not being igno...

```
str_replace_editor str_replace /testbed/reproduce_issue.py   --old_str '        # Test if files are
↪  correctly ignored
       print("
Testing file ignoring:")
       print(f"test.yaml ignored: \{config.is_file_ignored(os.path.join(tmpdir, '"'"'test....
```

---

**Environment**

```
The file /testbed/reproduce_issue.py has been edited. Here's the result of running `cat -n` on a snippet of
↪  /testbed/reproduce_issue.py:
   37            try:
   38                # Try to create a config with a list of i...
```

---

**Expert Model Agent**

Let's run the updated script:

```
python /testbed/reproduce_issue.py
```

---

---

**Environment**

```
Config created successfully!

Testing file ignoring:
Path: /tmp/tmp6ed2f08s/test.yaml
test.yaml ignored: False

Path: /tmp/tmp6ed2f08s/test.dont-lint-me.yaml
test.dont-lint-me.yaml ignored: True

Path: /tmp/tmp6ed2f08s/bin/test.yaml
bin...
```

**Expert Model Agent**

I see the issue now. The patterns are being correctly loaded, but the '/bin/' pattern is not matching the path '/tmp/tmp6ed2f08s/bin/test.yaml'. This is because the 'pathspec' library is matching against the full path, not just the relative path.
Le...

```
str_replace_editor str_replace /testbed/reproduce_issue.py   --old_str '      # Create test files
    with open(os.path.join(tmpdir, "test.yaml"), "w") as f:
        f.write("key: value
")

    with open(os.path.join(tmpdir, "test.dont-lint-me.yam...
```

**Environment**

```
The file /testbed/reproduce_issue.py has been edited. Here's the result of running `cat -n` on a snippet of
↪ /testbed/reproduce_issue.py:
    9
   10          from yamllint.config import YamlLintConfig
   11
   ...
```

**Expert Model Agent**

```
str_replace_editor str_replace /testbed/reproduce_issue.py   --old_str '          # Test if files are
↪    correctly ignored
        print("
Testing file ignoring:")
        test_yaml_path = os.path.join(tmpdir, '"'"'test.yaml'"'"')
        dont_lint_path ...
```

**Environment**

```
The file /testbed/reproduce_issue.py has been edited. Here's the result of running `cat -n` on a snippet of
↪ /testbed/reproduce_issue.py:
   43          try:
   44              # Try to create a config with a list o...
```

**Expert Model Agent**

```
str_replace_editor str_replace /testbed/reproduce_issue.py   --old_str '    except Exception as e:
        print(f"
Error with rule-specific config: \{e\}")' --new_str '    except Exception as e:
        print(f"
Error with rule-specific config: \{e\}"...
```

**Environment**

```
The file /testbed/reproduce_issue.py has been edited. Here's the result of running `cat -n` on a snippet of
↪ /testbed/reproduce_issue.py:
   83
   84          try:
   85              # Try to create a config with rule-specific ignore ...
```

**Expert Model Agent**

Let's run the updated script:

```
python /testbed/reproduce_issue.py
```

**Environment**

```
Config created successfully!

Testing file ignoring:
Path: test.yaml
test.yaml ignored: False

Path: test.dont-lint-me.yaml
test.dont-lint-me.yaml ignored: True

Path: bin/test.yaml
bin/test.yaml ignored: True

Path: bin/test.lint-me-an...
```

**Expert Model Agent**

Great! Now the ignore patterns are working correctly:
1. 'test.yaml' is not ignored (as expected) 2. 'test.dont-lint-me.yaml' is ignored (matching the pattern '*.dont-lint-me.yaml') 3. 'bin/test.yaml' is ignored (matching the pattern '/bin/') 4. 'bi...

```
str_replace_editor str_replace /testbed/reproduce_issue.py  --old_str '    try:
        # Try to create a config with rule-specific ignore patterns
        rule_config = YamlLintConfig(content=rule_config_content)
        print("
Rule-specific conf...
```

**Environment**

```
The file /testbed/reproduce_issue.py has been edited. Here's the result of running `cat -n` on a snippet of
↪ /testbed/reproduce_issue.py:
    79          ignore:
    80            - "ign-trail"
    81            - "...
```

**Expert Model Agent**

Let's run the updated script:

```
python /testbed/reproduce_issue.py
```

**Environment**

```
Config created successfully!

Testing file ignoring:
Path: test.yaml
test.yaml ignored: False

Path: test.dont-lint-me.yaml
test.dont-lint-me.yaml ignored: True

Path: bin/test.yaml
bin/test.yaml ignored: True

Path: bin/test.lint-me-an...
```

**Expert Model Agent**

It looks like the rule-specific ignore patterns are not working correctly. Let's check the 'enabled_rules' method in the 'YamlLintConfig' class:

```
str_replace_editor view /testbed/yamllint/config.py  --view_range 50 60
```

**Environment**

```
Here's the result of running `cat -n` on /testbed/yamllint/config.py:
    50          def is_yaml_file(self, filepath):
    51              return self.yaml_files.match_file(os.path.basename(filepath))
    52
    53          def enabled_rules(self, f...
```

**Expert Model Agent**

The issue is in the 'enabled_rules' method. It's checking if 'val['ignore'].match_file(filepath)' is true, but it's not handling the case where 'val['ignore']' is a list. Let's modify our test script to print more information about the rule-specific ...

```
str_replace_editor str_replace /testbed/reproduce_issue.py  --old_str '       # Test rule-specific ignore
        print("
Testing rule-specific ignore:")

        # Get the enabled rules for each file
        print("Enabled rules for ign-d...
```

**Environment**

```
The file /testbed/reproduce_issue.py has been edited. Here's the result of running `cat -n` on a snippet of
↪ /testbed/reproduce_issue.py:
    94                       f.write("key: value
")
    95
 ...
```

**Expert Model Agent**

Let's run the updated script:

```
python /testbed/reproduce_issue.py
```

**Environment**

```
Config created successfully!

Testing file ignoring:
Path: test.yaml
test.yaml ignored: False

Path: test.dont-lint-me.yaml
test.dont-lint-me.yaml ignored: True

Path: bin/test.yaml
bin/test.yaml ignored: True

Path: bin/test.lint-me-an...
```

**Expert Model Agent**

Now I see the issue. The rule-specific ignore patterns are not being converted to 'PathSpec' objects. Let's check the 'validate_rule_conf' function in the 'config.py' file:

```
str_replace_editor view /testbed/yamllint/config.py  --view_range 145 175
```

**Environment**

```
Here's the result of running `cat -n` on /testbed/yamllint/config.py:
   145                   raise YamlLintConfigError(f'invalid config: \{e\}') from e
   146
   147               self.rules[id] = validate_rule_conf(rule, self.rules[id])
 ...
```

**Expert Model Agent**

The issue is that our fix for the 'validate_rule_conf' function is correct, but the rule-specific ignore patterns are not being properly processed. Let's check if there's any other place in the code that might be affecting this.
Let's modify our tes...

```
str_replace_editor str_replace /testbed/reproduce_issue.py  --old_str '       # Print rule-specific ignore
↪ patterns
        print("Rule-specific ignore patterns:")
        for rule_id, rule_conf in rule_config.rules.items():
            if isinstan...
```

**Environment**

```
The file /testbed/reproduce_issue.py has been edited. Here's the result of running `cat -n` on a snippet of
↪  /testbed/reproduce_issue.py:
    97                    f.write("key: value
")
    98
    99               ...
```

**Expert Model Agent**

Let's run the updated script:

```
python /testbed/reproduce_issue.py
```

**Environment**

```
Config created successfully!

Testing file ignoring:
Path: test.yaml
test.yaml ignored: False

Path: test.dont-lint-me.yaml
test.dont-lint-me.yaml ignored: True

Path: bin/test.yaml
bin/test.yaml ignored: True

Path: bin/test.lint-me-an...
```

**Expert Model Agent**

Now I see the issue. The rule-specific ignore patterns are being correctly converted to 'PathSpec' objects, but they're not matching the files correctly. This is likely because the patterns are not being interpreted correctly.
Let's check the 'enabl...

```
str_replace_editor view /testbed/yamllint/config.py  --view_range 50 60
```

**Environment**

```
Here's the result of running `cat -n` on /testbed/yamllint/config.py:
    50          def is_yaml_file(self, filepath):
    51              return self.yaml_files.match_file(os.path.basename(filepath))
    52
    53          def enabled_rules(self, f...
```

**Expert Model Agent**

The issue is in the 'enabled_rules' method. It's using 'val['ignore'].match_file(filepath)' to check if a file should be ignored for a specific rule, but it's not considering the basename of the file like it does in the 'is_yaml_file' method.
Let's ...

```
str_replace_editor str_replace /testbed/reproduce_issue.py  --old_str '                     # Test if the ignore
↪  pattern works
                if rule_id == '"'"'key-duplicates'"'"':
                    print(f"    ign-dup.yaml ignored: \{rule_conf['"'"'i...
```

**Environment**

```
The file /testbed/reproduce_issue.py has been edited. Here's the result of running `cat -n` on a snippet of
↪  /testbed/reproduce_issue.py:
   109                      for pattern in rule_conf['ignore'].patterns:
   110              ...
```

**Expert Model Agent**

Let's run the updated script:

```
python /testbed/reproduce_issue.py
```

**Environment**

```
Config created successfully!

Testing file ignoring:
Path: test.yaml
test.yaml ignored: False

Path: test.dont-lint-me.yaml
test.dont-lint-me.yaml ignored: True

Path: bin/test.yaml
bin/test.yaml ignored: True

Path: bin/test.lint-me-an...
```

**Expert Model Agent**

Now I see the issue. The patterns are matching the files without the '.yaml' extension, but not with the extension. This is because the patterns in the config file are specified without the extension.
Let's modify the 'enabled_rules' method in the '...

```
str_replace_editor str_replace /testbed/yamllint/config.py  --old_str '    def enabled_rules(self,
↪  filepath):
        return [yamllint.rules.get(id) for id, val in self.rules.items()
                if val is not False and (
                    file...
```

**Environment**

```
The file /testbed/yamllint/config.py has been edited. Here's the result of running `cat -n` on a snippet of
↪  /testbed/yamllint/config.py:
    48                return self.ignore and self.ignore.match_file(filepath)
    49
    50        ...
```

**Expert Model Agent**

Let's run the script again to see if our changes fixed the issue:

```
python /testbed/reproduce_issue.py
```

**Environment**

```
Config created successfully!

Testing file ignoring:
Path: test.yaml
test.yaml ignored: False

Path: test.dont-lint-me.yaml
test.dont-lint-me.yaml ignored: True

Path: bin/test.yaml
bin/test.yaml ignored: True

Path: bin/test.lint-me-an...
```

**Expert Model Agent**

Great! Now the 'trailing-spaces' rule is being correctly disabled for the 'ign-trail.yaml' file, but not for the 'ign-trail.lint-me-anyway.yaml' file, which is what we want. However, the 'key-duplicates' rule is still not being disabled for the 'ign...

```
str_replace_editor str_replace /testbed/reproduce_issue.py  --old_str 'rule_config_content = """
extends: default
rules:
  key-duplicates:
    ignore:
      - "/ign-dup"
  trailing-spaces:
    ignore:
      - "ign-trail"
      - "!*.lint-me-anyway.y...
```

---

**Environment**

```
The file /testbed/reproduce_issue.py has been edited. Here's the result of running `cat -n` on a snippet of
↪ /testbed/reproduce_issue.py:
    67
    68              except Exception as e:
    69                  print(f"Error:...
```

---

**Expert Model Agent**

Let's run the script again:

```
python /testbed/reproduce_issue.py
```

---

**Environment**

```
Config created successfully!

Testing file ignoring:
Path: test.yaml
test.yaml ignored: False

Path: test.dont-lint-me.yaml
test.dont-lint-me.yaml ignored: True

Path: bin/test.yaml
bin/test.yaml ignored: True

Path: bin/test.lint-me-an...
```

---

**Expert Model Agent**

Now both rules are working correctly. The 'key-duplicates' rule is being disabled for the 'ign-dup.yaml' file, and the 'trailing-spaces' rule is being disabled for the 'ign-trail.yaml' file but not for the 'ign-trail.lint-me-anyway.yaml' file.
Let's...

```
str_replace_editor create /testbed/comprehensive_test.py --file_text '#!/usr/bin/env python3

import os
import tempfile
import sys

# Add the current directory to the Python path
sys.path.insert(0, os.path.abspath(os.path.dirname(__file__)))

from ya...
```

---

**Environment**

```
File created successfully at: /testbed/comprehensive_test.py
```

---

**Expert Model Agent**

Let's run the comprehensive test script:

```
python /testbed/comprehensive_test.py
```

---

**Environment**

```
Testing top-level ignore configuration...
test.yaml ignored: False
test.dont-lint-me.yaml ignored: True
test.yaml ignored: False
test.dont-lint-me.yaml ignored: True
bin/test.yaml ignored: True
bin/test.lint-me-anyway.yaml ignored: Fals...
```

---

**Expert Model Agent**

Great! The comprehensive test script shows that our changes are working correctly:
1. Both string and list formats for the top-level 'ignore' configuration work correctly. 2. Both string and list formats for rule-specific 'ignore' configurations wor...

```
cd /testbed \&\& python -m unittest discover
```

---

**Environment**

. . . . . . . . . . . . . . . . . . . . . . . . . . . . . . . . . . . . . . . . . . . . . . . . . . . . . . . . . . . . . . . . . . . . . . . . . . . . . . . . . . . . . . . . . . . . . . . . .

**Expert Model Agent**

All the tests pass, which means our changes don't break any existing functionality.
Let's summarize the changes we've made:
1. We modified the 'parse' method in the 'YamlLintConfig' class to accept both a string and a list for the top-level 'ignore...

```
submit
```

# B. Hyperparameters

| Hyperparameter | 7B Models | 32B Models |
|---|---|---|
| Batch size | 64 | 16 |
| Training epochs | 3 | 3 |
| Learning rate | $5 \times 10^{-5}$ | $5 \times 10^{-5}$ |
| Warmup steps | 5 | 5 |
| Maximum sequence length | 32768 | 32768 |
| Max grad norm | 1.0 | 1.0 |
| Optimizer | Adam | Adam |
| Adam $\beta_1$ | 0.9 | 0.9 |
| Adam $\beta_2$ | 0.999 | 0.999 |
| LR schedule | cosine | cosine |

*Table 5.* Hyperparameters for our 7B and 32B supervised fine-tuning experiments.

# C. Model Failure Analysis

## C.1. Judge Prompt

```
You are an expert software engineer analyzing why a software engineering agent failed to resolve an issue.

INSTANCE ID: {instance_id}
{exit_status_desc}

AVAILABLE AGENT ACTIONS:

---- BEGIN FUNCTION #1: bash ----
Description: Execute a bash command in the terminal.
* Can generate very large outputs when listing files (ls, find, grep)
* Output contributes directly to context window usage
* Commands like 'find /repo -name "*.py"' can list thousands of files
* Large outputs can quickly fill the context window

Parameters:
  (1) command (string, required): The bash command to execute. Can be empty to view additional logs when
  ↪  previous exit code is `-1`. Can be `ctrl+c` to interrupt the currently running process.
---- END FUNCTION #1 ----

---- BEGIN FUNCTION #2: submit ----
Description: Finish the interaction when the task is complete OR if the assistant cannot proceed further
↪  with the task.
* Used when agent thinks task is done (may be correct or incorrect solution)
* Also used when agent is stuck and cannot make progress
* No parameters are required for this function.
---- END FUNCTION #2 ----

---- BEGIN FUNCTION #3: str_replace_editor ----
Description: Custom editing tool for viewing, creating and editing files
* State is persistent across command calls and discussions with the user
* If `path` is a file, `view` displays the result of applying `cat -n`. If `path` is a directory, `view`
↪  lists non-hidden files and directories up to 2 levels deep
* Directory views can generate large outputs contributing to context usage
* The `create` command cannot be used if the specified `path` already exists as a file
* If a `command` generates a long output, it will be truncated and marked with `<response clipped>`
* The `undo_edit` command will revert the last edit made to the file at `path`

Notes for using the `str_replace` command:
* The `old_str` parameter should match EXACTLY one or more consecutive lines from the original file. Be
↪  mindful of whitespaces!
* If the `old_str` parameter is not unique in the file, the replacement will not be performed. Make sure to
↪  include enough context in `old_str` to make it unique
* The `new_str` parameter should contain the edited lines that should replace the `old_str`

Parameters:
  (1) command (string, required): The commands to run. Allowed options are: `view`, `create`, `str_replace`,
  ↪  `insert`, `undo_edit`.
  (2) path (string, required): Absolute path to file or directory, e.g. `/repo/file.py` or `/repo`.
  (3) file_text (string, optional): Required parameter of `create` command, with the content of the file to
  ↪  be created.
  (4) old_str (string, optional): Required parameter of `str_replace` command containing the string in `path`
  ↪  to replace.
  (5) new_str (string, optional): Optional parameter of `str_replace` command containing the new string (if
  ↪  not given, no string will be added). Required parameter of `insert` command containing the string to
  ↪  insert.
  (6) insert_line (integer, optional): Required parameter of `insert` command. The `new_str` will be inserted
  ↪  AFTER the line `insert_line` of `path`.
  (7) view_range (array, optional): Optional parameter of `view` command when `path` points to a file. If
  ↪  none is given, the full file is shown. If provided, the file will be shown in the indicated line
  ↪  number range, e.g. [11, 12] will show lines 11 and 12. Indexing at 1 to start. Setting `[start_line,
  ↪  -1]` shows all lines from `start_line` to the end of the file.
---- END FUNCTION #3 ----
```

```
---- BEGIN FUNCTION #4: file_viewer ----
Description: Interactive file viewer for opening and navigating files in the editor.
* open <path> [<line_number>]: Opens the file at path. If line_number is provided, the view moves to include
↪  that line.
* goto <line_number>: Moves the window to show the specified line number.
* scroll_down: Moves the window down 100 lines.
* scroll_up: Moves the window up 100 lines.

Parameters:
  (1) command (string, required): One of `open`, `goto`, `scroll_down`, `scroll_up`.
  (2) path_or_line (string/int, optional): For `open`, a path (and optional line). For `goto`, a line number.
---- END FUNCTION #4 ----

---- BEGIN FUNCTION #5: search_tools ----
Description: Searching utilities for locating text or files within the workspace.
* search_file <search_term> [<file>]: Searches for search_term in file. If file is not provided, searches
↪  the current open file.
* search_dir <search_term> [<dir>]: Searches for search_term in all files in dir. If dir is not provided,
↪  searches in the current directory.
* find_file <file_name> [<dir>]: Finds all files with the given name in dir. If dir is not provided,
↪  searches in the current directory.

Parameters:
  (1) subcommand (string, required): One of `search_file`, `search_dir`, `find_file`.
  (2) arg1 (string, required): The search term or file name, depending on subcommand.
  (3) arg2 (string, optional): Target file (for search_file) or directory (for search_dir/find_file).
---- END FUNCTION #5 ----

---- BEGIN FUNCTION #6: edit_block ----
Description: Block editor for replacing ranges in the current open file and finalizing edits.
* edit <n>:<m> <replacement_text>: Replaces lines n through m (inclusive) with the given text in the open
↪  file. Ensure indentation is correct.
* end_of_edit: Applies the pending changes. Python files are syntax-checked after the edit; if an error is
↪  found, the edit is rejected.

Parameters:
  (1) command (string, required): `edit` or `end_of_edit`.
  (2) range_and_text (varies): For `edit`, a line range `n:m` and the replacement text.
---- END FUNCTION #6 ----

---- BEGIN FUNCTION #7: create_file ----
Description: Creates and opens a new file with the given name.

Parameters:
  (1) filename (string, required): Absolute or workspace-relative path to create. The file must not already
  ↪  exist.
---- END FUNCTION #7 ----

PROBLEM STATEMENT:
{problem_statement}

FINAL ACTIONS TAKEN (Last {NUM_PAST_ACTIONS}):
{chr(10).join(final_actions[-NUM_PAST_ACTIONS:]) if final_actions else "No actions recorded"}

FINAL OBSERVATIONS (Last {NUM_PAST_ACTIONS}):
{chr(10).join(final_observations[-NUM_PAST_ACTIONS:]) if final_observations else "No observations recorded"}

TRAJECTORY SUMMARY:
- Total steps: {len(trajectory_steps)}
- Final state: Failed (no successful patch generated)

ANALYSIS INSTRUCTIONS:
The exit status indicates WHY the agent terminated. Consider how the final actions contributed to this
↪  specific exit condition.

Based on the information above, provide an error analysis in two parts:
First, an explanation of the issue and why the trajectory failed.
Second, a category for the error.

Wrap your explanation in <description></description> tags.
```

```
For the category, choose EXACTLY one from the following set: identified_incorrect_file: The agent
↪  incorrectly identified the file that needed to be fixed., missed_edge_case: The agent missed an edge
↪  case in one of the test cases., misunderstood_problem_statement: The agent misunderstood the problem
↪  statement., wrong_solution: The agent generated a wrong solution., tool_error: The agent encountered an
↪  error while using a tool (e.g. by calling it incorrectly)., infinite_loop: The agent entered an infinite
↪  loop (e.g. repeating the same sequence of steps)., endless_file_reading: The agent read the same file
↪  multiple times without making any changes., context_overflow_from_listing: The agent's file listing
↪  operations (ls, find, etc.) caused context overflow., syntax_error: The agent generated syntactically
↪  incorrect code., other: The agent failed to resolve the issue for other reasons.
Do NOT invent or propose new categories. If none fits, use "other".

Place the category at the end, separated by two newlines. Category must be
all lowercase and only list the category name.

Remember to write two new lines before the category.
```

## C.2. Sample Judgements

```
The task was to fix a bug where duration-only expressions (e.g., F('estimated_time') + timedelta) don't work
on SQLite and MySQL.
The agent correctly inferred that the fix likely involves backend-specific handling of duration expressions
or database converters (e.g., get_db_converters in mysql/sqlite operations).
They searched for get_db_converters and opened mysql/operations.py and sqlite3/operations.py multiple times.

However, the agent got stuck repeatedly viewing the same files with various view ranges, often resulting in
truncated and partial outputs (e.g., lines cut mid-word like "from django.db.backends.base.operat" and
"field_name").
They also attempted an invalid view range once, but generally kept re-opening the same files without making
any edits.
No attempt was made to modify code (e.g., overriding get_db_converters for DurationField or adding duration
conversion logic), and no patch was generated.

The failure occurred because the agent entered a loop of re-reading the same backend files without
performing any actual code changes.
The repeated file views and small tool missteps led to no progress on implementing the necessary fix,
resulting in termination with no patch produced.

Category: endless_file_reading
```

```
Issue and why the trajectory failed:
- The agent attempted to fix xr.where not preserving attributes by directly modifying
xarray/core/computation.py's where function. They added manual logic to copy attrs from x (and sometimes y),
update result.attrs after apply_ufunc, and even cast dtype back to x's dtype.
- Despite multiple large str_replace edits, the verification script continued to print that attributes were
empty and dtype changed (int8 -> int64), indicating the patch did not address the real cause. The function
already passed keep_attrs to apply_ufunc in this codebase, so the true fix likely required changes in
apply_ufunc's attribute handling or how "override/keep/drop" semantics are implemented, rather than ad-hoc
copying in where.
- The agent repeatedly edited the same block with bulky replacements and speculative logic (including
special cases for numpy arrays), but never validated whether the local xarray being imported was from
/testbed or an environment-installed version. If the latter, none of the edits would affect runtime
behavior, explaining why the test output didn't change.
- The approach diverged from the PR's intent (enabling attribute preservation via apply_ufunc/keep_attrs),
adding unrelated dtype casting and brittle attribute logic. No unit tests were added or existing tests run
in a way to ensure the modified code path was actually used.

In short, the agent implemented a wrong fix in the wrong place and didn't ensure the edits were executed, so
the issue persisted and no successful patch was produced.

Category: wrong_solution
```

## D. Student-Expert Covariate Shift KL-Divergence

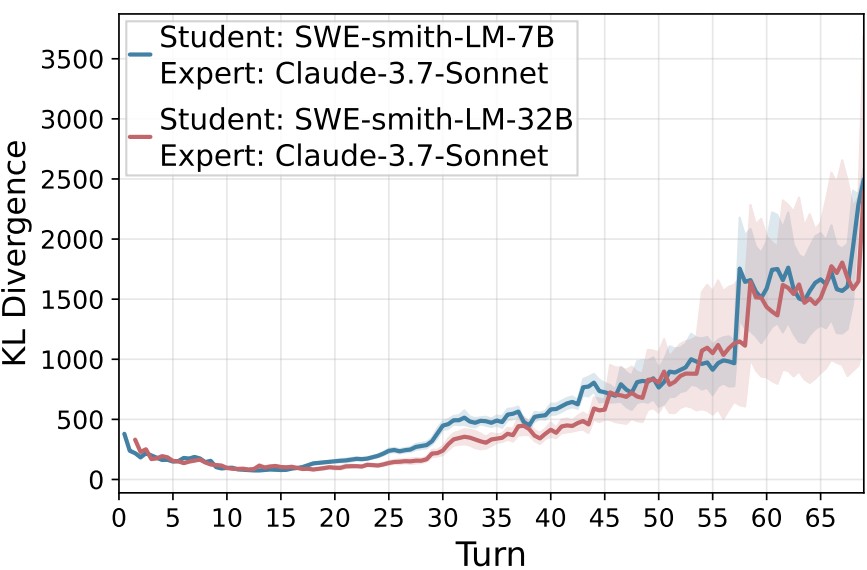

*Figure 5.* Covariate shift, as measured by KL Divergence, between student and expert throughout a set of trajectories; analogous to the Fréchet Distance (FD), distributional divergence increases over turns.

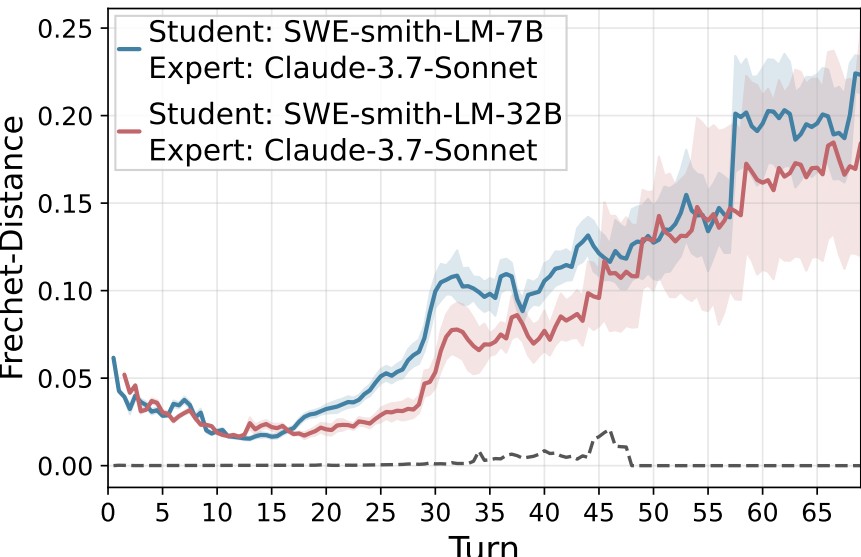

*Figure 6.* Covariate shift between student and expert throughout a set of trajectories, with a dashed baseline of FD between an expert-expert partition.

