# OpenReview forum: "Imitation Learning for Multi-turn LM Agents via On-policy Expert Corrections"
_ICML.cc/2026/Conference — ICML 2026 regular_

### Official Review · Reviewer_A2pJ · 2026-02-28

**Soundness:** 3
**Presentation:** 3
**Significance:** 3
**Originality:** 3
**Overall Recommendation:** 4
**Confidence:** 4

**Summary:**

This paper addresses covariate shift in multi-turn language model agent training by introducing on-policy expert corrections (OECs): trajectories that begin on-policy with a student model and switch mid-trajectory to an expert, allowing the expert to complete the task and enabling verifiable-reward filtering. The method adapts the roll-in/roll-out spirit of DAgger to the LM agent setting and is evaluated on software engineering tasks using SWE-agent scaffolds, with additional masking of on-policy tokens and automated filtering of repetitive failure modes. Experiments with Qwen2.5-Coder 7B and 32B show consistent gains over standard behavioral cloning and fully on-policy trajectories, along with an embedding-based analysis showing covariate shift grows across turns.

**Compliance With Llm Reviewing Policy:**

Affirmed.

**Final Justification:**

I maintain my final recommendation at 4 (Weak Accept). I find the paper technically solid, practically relevant, and well presented, with a clear empirical contribution on addressing covariate shift in multi-turn LM agent training. The main strengths are soundness, clarity, and practical significance; originality is moderate, since the method is best viewed as a pragmatic adaptation of DAgger-style ideas to the LM-agent setting rather than a fundamentally new algorithm. The rebuttal addressed my main concerns reasonably well: it clarified the “on-policy” issue, added a stronger RLVR comparison, and, importantly, provided the requested quantitative token-efficiency analysis to better support the cost/performance trade-off discussion. Remaining limitations, such as heuristic switch policies, lack of multi-seed variance, and limited analysis of repository-edit corrections, still constrain the paper’s impact, but they do not outweigh the overall strength of the empirical results and practical value. Overall, the rebuttal reinforced my prior positive assessment rather than materially changing it.

**Key Questions For Authors:**

* Why choose a uniform switch distribution over 0-30? How does performance vary with different switch distributions or with multiple OEC rounds that progressively refresh on-policy states as the student improves?
* The authors should report the training and evaluation compute (e.g., number of tokens processed) for OEC, BC, and on‑policy trajectories, which is needed to substantiate the efficiency claims and to quantify the cost/performance trade‑offs.
* Could authors include stronger baselines under same scaffold (e.g., an RLVR baseline, or a one-step DAgger-style query baseline) to better contextualize OEC’s gains?
* How robust are the main results across multiple random seeds? The authors should add confidence intervals for the resolution rates in some experiments.

**Limitations:**

Yes

**Strengths And Weaknesses:**

Strengths:
* This paper adapts DAgger’s roll-in/roll-out concept to long-horizon LM agents by switching from a student rollout to an expert mid-trajectory and letting the expert complete the episode to retain verifiable reward signals.
* The experimental results show consistent gains in both the 7B and 32B regimes. Scaling curves plotted against data volume clearly illustrate improvements, and an additional analysis of the mixed OEC/BC dataset further validates the findings.
* It addresses a well-recognized and practically important challenge, i.e., covariate shift, in long-horizon LM agents, which is a growing area in machine learning.

Weaknesses:
* The “on-policy” claim is weakened by the necessity to rewrite the history into the expert’s scaffold/prompt format at the switch. This token-level distributional change could partially undermine the intended on-policy conditioning for the expert.
* The switch-time policy (uniform over 0–30 steps) is heuristic.
* Comparisons to RLVR under the same scaffold and compute budget are absent. And statistical uncertainty (e.g., multiple seeds) is not reported.
* There's limited analysis of whether experts should revert student-applied edits to the repo and how often this occurs or affects success.
* The tables should be formatted using the three‑line style, which is more formal and visually appealing. Moreover, some table references mistakenly use a “Figure” prefix (e.g., in 4.2 results), which should be corrected.

---

> ### Author Rebuttal · Authors · 2026-03-31
>
> Thank you for your review. We appreciate you noting that the “results show consistent gains” and that the paper “addresses a well-recognized and practically important challenge”. We address each point individually:
>
> > The “on-policy” claim is weakened by the necessity to rewrite the history into the expert’s scaffold/prompt….
>
> The “rewriting” of the history into the expert’s scaffold and prompt does not actually change the contents of the trajectory. It simply rewrites the trajectory into the format and initial prompt to match the formatting that the expert model expects. Functionally, the trajectory is otherwise exactly the same.
>
> > The switch-time policy (uniform over 0–30 steps) is heuristic...?
>
> We primarily chose a uniform distribution for simplicity and straightforward experimentation. This sampling strategy also allowed us to easily reuse (reuse is important since generating expert trajectories is very monetarily expensive) the samples to perform the “switch time” ablation to study its impact. We don’t claim that uniform sampling is the best strategy and see the **flexibility to easily implement new sampling strategies as one of OEC’s strengths**. We are excited to see more sophisticated sampling strategies developed in future work.
>
> > Comparisons to RLVR under the same scaffold and compute budget are absent.
>
> We **have run new experiments against a new recent RL approach** for finetuning LM agents in the SWE agent domain. We chose Agent-RLVR [1], a recent RL-based method for training LM agents in the SWE agent domain, for this comparison. So far we have tested Agent-RLVR under the same conditions presented in Figure 2a in the paper: the 7B student with rollouts on the same problem distribution and 2000 training samples. The eval results on SWE-bench Lite are presented in the following table and show that OEC achieves similar performance gains over Agent-RLVR:
>
>
> | Number of Samples | OEC (ours) | Agent-RLVR | On-policy | BC    |
> | ----------------- | ---------- | ---------- | --------- | ----- |
> | 2000              | **14.7%**    | 13.7%      | 13.7%     | 13.7% |
>
> So far, our results indicate that Agent-RLVR achieves similar performance as OEC and on-policy in the 7B setting. We thank the reviewers for this suggestion since we believe a direct comparison against an RL-based method greatly strengthens the results and claims of the paper, demonstrating the OECs shows improvement over both traditional imitation learning and RL.
>
> [1] Agent-RLVR: Training Software Engineering Agents via Guidance and Environment Rewards, J Da et al. 2025.
>
> > And statistical uncertainty (e.g., multiple seeds) is not reported. How robust are the main results across multiple random seeds?...
>
> We use a temperature of zero for our rollout generation and evaluation so there is no variance from these factors. Technically there can be some variance from the SFT training since the order of data is randomized, but we use the standard approach (due to computational limits) of running experiments over a single SFT training run for each datapoint. We hope that by providing **several data points across model sizes (7B and 32B), data scales (1000-4000 samples), multiple baselines (on-policy and BC), and ablations**, we’ve convincingly demonstrated an improvement over baselines.
>
> > There's limited analysis of whether experts should revert student-applied edits to the repo and how often this occurs or affects success.
>
> We provide some **qualitative analysis of different training methods in Table 3**, but investigating these fine-grained properties of why OECs trajectories work would be an exciting future direction.
>
> > The tables should be formatted using the three‑line style, which is more formal and visually appealing. … table references mistakenly use a “Figure” prefix (e.g., in 4.2 results)...
>
> Thank you for the suggestions, we will apply these fixes to the final version.
>
> > How does performance vary with multiple OEC rounds that progressively refresh on-policy states as the student improves?
>
> Collecting any kind of expert trajectory, including OECs, is financially expensive, so we did not explore experiments over multiple rounds of OEC trajectories. This would be an exciting direction for future work.
>
> > The authors should report the training and evaluation compute (e.g., number of tokens processed) for OEC, BC, and on‑policy trajectories…
>
> We agree that controlling for expert compute is important. We note, however, that behavioral cloning (BC) (the current most popular baseline used to fine-tune SWE agents) relies on trajectories that are fully generated by the expert, and therefore requires more expert queries overall than OEC, which only uses the expert for suffixes. To make this comparison more precise, **we will add a breakdown of student vs. expert token usage** for each method in the experiments.
>
> > Could authors include stronger baselines under same scaffold…?
>
> Yes, see our new results above.

---

> > ### Author Rebuttal · Reviewer_A2pJ · 2026-04-01
> >
> > Thank you for the detailed rebuttal. One specific piece of requested empirical evidence is still pending:
> > * While you logically explained that BC requires more expert queries than OEC and promised to add a token breakdown in the final paper, the actual token counts/compute metrics (student vs. expert) were not provided in the rebuttal.
> >
> > Could you please provide the estimated numbers or a rough quantitative breakdown of the token usage? This quantitative comparison is necessary to fully substantiate the cost/efficiency trade-off claims before I finalize my assessment.

---

> > > ### Author Response · Authors · 2026-04-01
> > >
> > > We have run the analysis that you requested and have some new results to share that we will also include in the final version of the paper.
> > >
> > > We analyzed the distribution of the number of expert and student tokens in the OEC trajectories and the distribution of the number of expert tokens required for the behavioral cloning (BC) baseline in the 32B setting (i.e., Figure 2B). Here are some summary statistics for the two distributions:
> > >
> > > | Method          | Mean  | Median |
> > > |-----------------|-------|--------|
> > > | BC (expert)     | 22812 | 18585  |
> > > | OEC (expert)    | 21116 | 18014  |
> > > | OEC (student)   | 5444  | 4372   |
> > >
> > > The results indicate that on average, OECs save 1696 expert tokens (~8%) over the traditional BC approach, aided by the property that OECs trajectories are partially generated via the student model. The average number of student tokens spent is 5444, indicating that the total number of tokens in OEC trajectories tends to be longer, but the number of expert tokens required is reduced. We will include these statistics and a histogram of the distribution in the final version of the paper. Thank you for suggesting running this analysis since this highlights an efficiency benefit of OECs over BC trajectories that we hadn’t considered and the analysis will strengthen the paper.

---

### Official Review · Reviewer_D16B · 2026-03-05

**Soundness:** 2
**Presentation:** 3
**Significance:** 3
**Originality:** 2
**Overall Recommendation:** 4
**Confidence:** 4

**Summary:**

The paper addresses covariate shift in imitation learning for multi-turn LM agents, by introducing On-policy Expert Corrections (OECs) as a data generation technique. OECs generate partially on-policy trajectories, by switching from a student model to an expert model partway through a rollout. The method is evaluated on SWE-benchmark.

**Compliance With Llm Reviewing Policy:**

Affirmed.

**Final Justification:**

While the rebuttal ameliorated some of my prior considerations, I still maintain that alternative exploration strategies could in theory match or exceed OECs gains. The paper does stand on its own as a worthy contribution, but could use more evaluations of alternative methods (DAGGER + other exploration policies?).

**Key Questions For Authors:**

Have you thought of more nuanced switch decision points? From your results I get that later switching helps more. Perhaps divergence-triggered switching (switch when off path), or even an exponential distribution (over duration).

Also, could it be possible to train on the on-policy steps as well? Ignoring them completely cannot be optimal. As it stands, it appears that an exploratory policy is all that's needed to lead the expert to unobserved states.

**Limitations:**

The paper honestly addresses most of its limitations. It could further explore the sensitivity to the switch index distribution. Closed source models also limit reproducibility.

**Strengths And Weaknesses:**

Soundness: I would say the empirical methodology of the paper is sound. There are minor objections on theoretical elements (mainly the lack of theoretical elements), particularly with respect to DAgger. The authors claim to invoke DAgger, but their method is substantially different to the point that no theoretical guarantees are preserved (they do mention that). Yet without DAgger, the work has no theoretical grounding (not a dealbreaker, just mentioning it). I also found it odd that they used a uniform distribution to select the switch-to-expert point.

Presentation: Very well presented. It's very clear what problem they address and how they approach it. I also admire the honesty of the claims. At no point were the authors overselling their (empirical) contributions.

Significance: Covariate shift poses a significant challenge in LLM fine tuning. The paper's OEC framework contributes to the effort of overcoming this challenge.

Originality: To my knowledge measuring the covariate shift is new. While DAgger is a known method, its application to LLMs is also novel, albeit a combination of existing ideas (yet if I hold the departure from DAgger and its theoretical guarantees as a slight on soundness, I must also report it a genuine original idea).

---

> ### Author Rebuttal · Authors · 2026-03-31
>
> Thank you for your review. We appreciate you noting that the “empirical methodology of the paper is sound”, that OECs are a “genuine original idea” and that the idea of “measuring the covariate shift is new.” We address each point individually:
>
> > I also found it odd that they used a uniform distribution to select the switch-to-expert point. Have you thought of more nuanced switch decision points? From your results I get that later switching helps more. Perhaps divergence-triggered switching (switch when off path), or even an exponential distribution (over duration).
>
> We primarily chose a uniform distribution for simplicity and straightforward experimentation. This sampling strategy also allowed us to easily reuse (reuse is important since generating expert trajectories is very monetarily expensive) the samples to perform the “switch time” ablation to study its impact. We don’t claim that uniform sampling is the best strategy and see the flexibility to easily implement new sampling strategies as one of OEC’s strengths. We are excited to see more sophisticated sampling strategies developed in future work.
>
> > Also, could it be possible to train on the on-policy steps as well? Ignoring them completely cannot be optimal.
>
> We investigated this exact question in our ablations and found that **training on on-policy (student) steps degrades performance**, particularly in the 32B setting. As shown in Table 2, removing on-policy masking largely eliminates the gains from OEC, indicating that including student actions introduces harmful overfitting. We believe this is due to compounding errors and low-quality behaviors in on-policy trajectories. We will clarify this point to emphasize that masking is an important design choice supported by our empirical results.
>
> > As it stands, it appears that an exploratory policy is all that's needed to lead the expert to unobserved states.
>
> We agree that exploration is important, but emphasize that the key requirement is not just exploration, but **on-policy state coverage**. This is a central insight from DAgger: the expert must be queried on the state distribution induced by the learned policy, not just arbitrary exploratory states.OEC follows this principle by using the student to generate on-policy prefixes, ensuring that expert corrections are conditioned on realistic failure modes of the student. This is more targeted than generic exploration and is a core motivation behind our design. We will better motivate this in the introduction of the paper.
>
> > Closed source models also limit reproducibility.
>
> Our **datasets, models, and code are already open-sourced**. They are just not included in the submission due to anonymity.
>
> We have also **run new experiments against a new recent RL approach** for finetuning LM agents in the SWE agent domain. We chose Agent-RLVR, a recent RL-based method for training LM agents in the SWE agent domain, for this comparison. So far we have tested Agent-RLVR under the same conditions presented in Figure 2a in the paper: the 7B student with rollouts on the same problem distribution and 2000 training samples. The eval results on SWE-bench Lite are presented in the following table and show that OEC achieves similar performance gains over Agent-RLVR:
>
> | Number of Samples | OEC (ours) | Agent-RLVR | On-policy | BC    |
> | ----------------- | ---------- | ---------- | --------- | ----- |
> | 2000              | **14.7%**    | 13.7%      | 13.7%     | 13.7% |

---

> > ### Author Rebuttal · Reviewer_D16B · 2026-04-01
> >
> > My comment on training on on-policy steps is fully resolved. The over-fitting problem is obvious in hindsight and the ablation study confirms this.
> >
> > My other 2 comments on uniform departure and exploration are related and still only partially resolved. Uniform departures do not guarantee that the expert will be asked to explore realistic failure modes of the student. OEC basically uses a random number of student policy steps as an exploration policy. Superior exploration policies could be an area for future work.

---

### Official Review · Reviewer_GFRm · 2026-03-11

**Soundness:** 3
**Presentation:** 3
**Significance:** 3
**Originality:** 3
**Overall Recommendation:** 4
**Confidence:** 3

**Summary:**

This paper studies the covariate shift problem in multi-turn LM agents trained with imitation learning, using SWE-agent-style software engineering tasks as the primary testbed. The authors propose OEC that rolls in with the student learning for a random number of turns and then switches to an expert agent to finish the trajectory, keeping only successful trajectories via unit-test rejection sampling, and fine-tuning while making the student prefix. The paper argues OEC is practical, LM-agent-friendly adaptation of Dagger-like ideas, and provides scaling curves and ablations showing OEC improves SWE-bench resolution rates.

**Compliance With Llm Reviewing Policy:**

Affirmed.

**Final Justification:**

Thank you to the authors for the detailed rebuttal and follow-up clarifications. Overall, my concerns are fully resolved: the response clearly positions OEC as a targeted data-generation scheme to mitigate covariate shift in multi-turn LM agents, and it strengthens the empirical story by clarifying the experimental setup (including confirmation that the same scaffold and filtering are used across baselines) and providing additional evidence such as compute/token usage breakdown and an RL-based comparison under matched conditions. While I still view the contribution primarily as a pragmatic adaptation of DAgger-style ideas to LM-agent settings rather than a fundamentally new algorithm, the rebuttal sufficiently addresses my key questions about attribution and fairness of comparisons. The remaining limitations (e.g., adaptive switching policies and stronger robustness/variance reporting) are better framed as future work and do not block acceptance.

**Key Questions For Authors:**

Please see weakness.

**Limitations:**

yes

**Strengths And Weaknesses:**

Strength:

1. OEC is easy to implement in an LM-agent scaffold, and the paper’s value is in the LM-specific design choices such as masking the student prefix, filtering loops/repetitions, and combining this with verification/rejection sampling.

2. The ablations are genuinely useful because they show which implementation designs actually matter.

3. Performance is competitive on SWE-bench.

Weakness:

1. Overall, I think the proposed approach might be viewed as a pragmatic adaptation of DAgger-style roll-in/roll-out data aggregation to multi-turn LM agents, combined with several important implementation choices. While effective, the algorithmic novelty beyond prior imitation-learning formulations appears limited; much of the gain may stem from a well-engineered data-generation and filtering pipeline tailored to non-Markovian LM agent trajectories. I think the paper would be strengthened by a DAgger-like baseline under the same scaffold and verification filtering. This would isolate whether the primary gains come from the OEC or from LM-specific heuristics such as prefix masking and repetition filtering.

2. Since OEC requires an expert to generate suffixes, it would be much more convincing to report results under a fixed budget. Without budget-matched comparisons, it is hard to tell whether OEC is strictly more sample-efficient or simply spending more expert compute in a different way.

3. Uniform random switching is simple, but it may be suboptimal because the best time to intervene likely depends on whether the student is stuck or uncertain. The manuscript provides some evidence that switch timing matters, but it does not propose or test an adaptive switching rule.

4. Some comparisons are relatively close, and the paper would benefit from reporting variance or confidence intervals, especially for key ablations and scaling trends.

---

> ### Author Rebuttal · Authors · 2026-03-31
>
> Thank you for your review. We appreciate your pointing out the “several important implementation choices” we made in adapting DAgger to a LM setting and noting that our “ablations are genuinely useful.” We address each point individually:
>
> > 1. Overall, I think the proposed approach might be viewed as a pragmatic adaptation of DAgger-style roll-in/roll-out data aggregation to multi-turn LM agents … non-Markovian LM agent trajectories.
>
> OEC is not simply a data filtering pipeline, but a new data generation scheme that produces partially on-policy, expert-completed trajectories. Prior imitation learning approaches in this setting (e.g., SWE-Smith) rely purely on off-policy expert rollouts, whereas OEC explicitly targets covariate shift by conditioning expert actions on on-policy states. To the best of the author’s knowledge, we are the first to explore extending DAgger to a language model setting.
>
> Moreover, **all baselines in our paper (BC and on-policy trajectories) use the same scaffold, rejection sampling, and filtering pipeline**. Thus, the performance gains isolate the effect of OEC-style data generation rather than the engineering components.
>
> > I think the paper would be strengthened by a DAgger-like baseline … and repetition filtering.
>
> A faithful DAgger baseline is difficult to implement in this setting: it requires querying the expert at many intermediate steps, which is prohibitively expensive for long-horizon LM trajectories and prevents reuse of KV caches (leading to O(T^2) cost). Our early experiments showed **traditional DAgger to be infeasible to compute even in the 7B setting**. It also does not naturally support trajectory-level verification (e.g., rejection sampling), which is central to our setup. We are not necessarily arguing the OEC is *better* than traditional DAgger, we simply adapt DAgger in a way that makes it practical to use in a language model setting. We will clarify these points in the revision of the paper.
>
> > 2. Since OEC requires an expert to generate suffixes, it would be much more convincing to report results under a fixed budget. Without budget-matched comparisons, it is hard to tell whether OEC is strictly more sample-efficient or simply spending more expert compute in a different way.
>
> We agree that controlling for expert compute is important. We note, however, that behavioral cloning (BC) (the current most popular baseline used to fine-tune SWE agents) relies on trajectories that are fully generated by the expert, and therefore requires more expert queries overall than OEC, which only uses the expert for suffixes. To make this comparison more precise, **we will add a breakdown of student vs. expert token usage** for each method in the experiments.
>
> > 3. Uniform random switching is simple, but it may be suboptimal … adaptive switching rule.
>
> We agree that adaptive switching is a promising direction. In practice, however, reliably detecting when the student is “stuck” is non-trivial in this setting. Simple heuristics (e.g., repeated actions) can be unreliable due to environment variability, while more robust approaches (e.g., LLM-based judges) introduce significant additional compute. We view it as a strength that **uniform random switching already performs well, providing a simple and effective baseline without additional complexity**. We will clarify this discussion and highlight adaptive switching as an important direction for future work.
>
> > 4. Some comparisons are relatively close, and the paper would benefit from reporting variance or confidence intervals, especially for key ablations and scaling trends.
>
> We use a temperature of zero for our rollout generation and evaluation so there is no variance from these factors. Technically there can be some variance from the SFT training since the order of data is randomized, but we use the standard approach (due to computational limits) of running experiments over a single SFT training run for each datapoint. We hope that by providing several data points across model sizes (7B and 32B), data scales (1000-4000 samples), multiple baselines (on-policy and BC), and ablations, we’ve convincingly demonstrated an improvement over baselines.
>
> We have also run **new experiments against a new recent RL approach** for finetuning LM agents in the SWE agent domain. We chose Agent-RLVR, a recent RL-based method for training LM agents in the SWE agent domain, for this comparison. So far we have tested Agent-RLVR under the same conditions presented in Figure 2a with 2000 training samples. The eval results on SWE-bench Lite are presented in the following table and show that **OEC achieves similar performance gains over Agent-RLVR**:
>
> | Number of Samples | OEC (ours) | Agent-RLVR | On-policy | BC    |
> | ----------------- | ---------- | ---------- | --------- | ----- |
> | 2000              | **14.7%**    | 13.7%      | 13.7%     | 13.7% |

---

> > ### Author Rebuttal · Reviewer_GFRm · 2026-04-01
> >
> > Thank the authors for the detailed response. However, my core concern that the gains may largely come from LM-specific heuristics and that the algorithmic novelty vs DAgger is not cleanly isolated still isn’t fully resolved by the rebuttal alone. Also, the single-run / no-variance issue is not truly addressed (temp=0 doesn’t remove SFT training variability). For this, I have below follow up questions:
> > - Can you provide a budget-matched comparison (expert tokens / wall-clock) across BC vs OEC vs on-policy?
> > - Can you implement a cheap DAgger-like approximation (e.g., label only a subset of steps / shorter horizon) to better isolate novelty?
> > - Are masking/repetition filtering applied consistently across baselines, and how much of the gain comes from each component?
> > - Any evidence of robustness beyond a single SFT run (even 2–3 seeds for key points)?
> >
> > ===========
> >
> > Follow-up: Thank you to the authors for providing further details. I mark my concerns as fully resolved. The rebuttal clarifies OEC as a targeted data-generation scheme for addressing covariate shift in multi-turn LM agents and provides additional supporting details. While I still view the contribution as a pragmatic LM-agent adaptation of DAgger-style ideas rather than a fundamentally new algorithm, the remaining limitations (e.g., adaptive switching and stronger robustness/variance reporting) are best framed as future work and do not block acceptance.

---

> > > ### Author Response · Authors · 2026-04-01
> > >
> > > Thank you for your follow up questions. We hope our responses clarify your remaining concerns:
> > >
> > > > Can you provide a budget-matched comparison (expert tokens / wall-clock) across BC vs OEC vs on-policy?
> > >
> > > We analyzed the distribution of the number of expert and student tokens in the OEC trajectories and the distribution of the number of expert tokens required for the behavioral cloning (BC) baseline in the 32B setting (i.e., Figure 2B). Here are some summary statistics for the two distributions:
> > >
> > > | Method          | Mean  | Median |
> > > |-----------------|-------|--------|
> > > | BC (expert)     | 22812 | 18585  |
> > > | OEC (expert)    | 21116 | 18014  |
> > > | OEC (student)   | 5444  | 4372   |
> > >
> > > The results indicate that on average, OECs save 1696 expert tokens (~8%) over the traditional BC approach, aided by the property that OECs trajectories are partially generated via the student model. The average number of student tokens spent is 5444, indicating that the total number of tokens in OEC trajectories tends to be longer, but the number of expert tokens required is reduced. We will include these statistics and a histogram of the distribution in the final version of the paper. Thank you for suggesting running this analysis since this highlights an efficiency benefit of OECs over BC trajectories (albeit a minor one) that we hadn’t considered and the analysis will strengthen the paper.
> > >
> > > > Can you implement a cheap DAgger-like approximation (e.g., label only a subset of steps / shorter horizon) to better isolate novelty?
> > >
> > > Our paper is not trying to claim that OECs are better than other DAgger-like approaches but rather introducing the first DAgger-like algorithm to the LM agent setting. Unfortunately, since there are no other existing DAgger-like approximations in the LM agent setting, there isn’t enough time to implement and run such an experiment in the length of the rebuttal period. The primary focus of our work is around comparing OECs against existing methods for fine-tuning LM agents. We hope you can recognize the novelty of OECs as a first proposal for adapting DAgger to the LM agent setting and we hope future work will continue exploring new and better methods for adapting DAgger.
> > >
> > > > Are masking/repetition filtering applied consistently across baselines, and how much of the gain comes from each component?
> > >
> > > Yes, masking and repetition filtering is applied across all baselines. We will reiterate this point under the “Method” paragraph in section 4.1. Table 2 contains results for ablating masking and repetition filtering (demonstrating that both components are important) but we also copy it here for convenience:
> > >
> > > | Student model | No training | With OEC | No masking | No filtering |
> > > |---------------|------------|----------|------------|--------------|
> > > | Student-7b    | 13.3%      | 17%      | 17%        | 11%          |
> > > | Student-32b   | 31.2%      | 40%      | 31.8%      | 36.4%        |
> > >
> > > > Any evidence of robustness beyond a single SFT run (even 2–3 seeds for key points)?
> > >
> > > Our main experiments are replicated across several dataset sizes (1000 - 4000 samples, 4 - 7 data points for each method) and two model sizes (7B and 32B) and compared against multiple baselines (BC, on-policy, and now a new RLVR baseline). Along with the extra experiments (varying OEC and BC mixtures) and ablations (masking, filtering), we hope the number of data points is sufficient to support the main claims and demonstrate robustness.

---

### Official Review · Reviewer_RnQB · 2026-03-13

**Soundness:** 3
**Presentation:** 3
**Significance:** 1
**Originality:** 3
**Overall Recommendation:** 5
**Confidence:** 3

**Summary:**

This paper introduces the OECs (on-policy expert corrections) dataset, which aims to address the state distribution mismatch between the student policy and the expert policy as the number of turns increases in multi-turn LM agents, thereby improving the performance of fine-tuning on software engineering (SWE) tasks. The paper also proposes a quantitative analysis to measure the covariate shift between the student and expert models. The proposed approach is inspired by the switching-to-expert mechanism in DAgger, and is adapted to the specific characteristics of large language models (LLMs): the method introduces strategies such as rewriting the context and filtering repetitive trajectories. Experimental results show that OECs can improve the expert sample efficiency and achieve better performance.

**Compliance With Llm Reviewing Policy:**

Affirmed.

**Final Justification:**

The rebuttal has addressed my main concerns and I maintain my previous positive rating.

**Key Questions For Authors:**

(1) I am wondering why not first generate the complete responses and then query the expert for certain steps within the process? This approach seems more capable of satisfying the on-policy property.

(2) The OEC data consist of erroneous trajectories generated by the student model along with corrections provided by the expert model, so the dataset may still contain some incorrect instances. Could the authors provide a more detailed analysis of the proportion of expert model interventions in the data?

**Limitations:**

Yes

**Strengths And Weaknesses:**

**Strengths:**

The paper is logically rigorous, with a clear description of the algorithmic procedure, and the experimental section is comprehensive and reliable. In addition to evaluating the final performance and comparing with current state-of-the-art methods, the paper also quantitatively demonstrates through experiments the covariate shift phenomenon between the student and teacher as the number of turns increases, which is consistent with the earlier analysis. The authors further design a series of ablation studies to examine different components of the method, such as the ratio between BC samples and OEC samples, on-policy masking, repetition filtering, and the index for switching to the expert. Moreover, the authors employ LLM-as-a-judge to conduct a detailed analysis of the causes of errors. Overall, the work provides useful insights and reference value for this research area.


**Weaknesses:**


The paper acknowledges that OEC trajectories cannot inherit the no-regret learning guarantees of the traditional DAgger, but it does not provide theoretical or empirical analysis on how this deviation affects performance.

The paper’s use of the terms on-policy and off-policy appears somewhat inappropriate. In practice, the distinction reflected in the data is closer to that between online and offline data collection. Furthermore, in the experiments the authors use on-policy data (i.e., experiences generated by the model itself) as a training baseline. However, this setup is likely to lead to overfitting, and therefore provides limited value as a meaningful baseline.

---

> ### Author Rebuttal · Authors · 2026-03-31
>
> Thank you for your review. We appreciate your noting that our paper is “logically rigorous” and that the “experimental section is comprehensive and reliable.” We address each point individually:
>
> > The paper’s use of the terms on-policy and off-policy appears somewhat inappropriate. In practice, the distinction reflected in the data is closer to that between online and offline data collection.
>
> We agree that our use of on-policy vs off-policy and online vs offline could use some clarity, especially since the distinction of on-policy and off-policy often refers to RL methods. Specifically, our primary baseline (behavioral cloning) that is most prevalently used by existing works is off-policy data (because the data is generated via a policy different from the student) as well as being offline (a fixed expert dataset). Meanwhile, OECs are a mix of on-policy (the student roll-in) and off-policy (the expert roll-out) as well as being online (data is collected via the current policy). **We will include a full discussion of this in the introduction as well as in section 3** as well as being more precise about when we describe methods along the axis of on-policy vs off-policy and when we use the axis of online vs offline.
>
> > in the experiments the authors use on-policy data (i.e., experiences generated by the model itself) as a training baseline. However, this setup is likely to lead to overfitting, and therefore provides limited value as a meaningful baseline.
>
> We agree that the on-policy baseline can lead to overfitting and our qualitative results indicate that this is likely the case. However, the on-policy baseline is a popular recent approach for fine-tuning LMs on agentic trajectories and is analogous to using vanilla REINFORCE RL with the zero-one reward from unit tests. **We have also run new experiments against a new recent RL approach** for finetuning agents in the SWE agent domain. We chose Agent-RLVR, a recent RL-based method for training LM agents in the SWE agent domain, for this comparison. So far we have tested Agent-RLVR under the same conditions presented in Figure 2a in the paper with 2000 training samples. The eval results on SWE-bench Lite are presented in the following table and show that OEC achieves similar performance gains over Agent-RLVR:
>
> | Number of Samples | OEC (ours) | Agent-RLVR | On-policy | BC    |
> | ----------------- | ---------- | ---------- | --------- | ----- |
> | 2000              | **14.7%**    | 13.7%      | 13.7%     | 13.7% |
>
>
> > (1) I am wondering why not first generate the complete responses and then query the expert for certain steps within the process? This approach seems more capable of satisfying the on-policy property.
>
> The proposed approach—first generating full student trajectories and then querying the expert at intermediate steps—seems analogous to the original DAgger formulation (Ross et al., 2011), which we discuss in Section 3.1 (“Comparison to DAgger”). We will clarify this comparison in the revision and better highlight the main two practical considerations that motivated our design:
> 1. *Complexity*: In standard DAgger-style querying, the expert must be invoked at many intermediate steps, **requiring recomputation of the KV cache from scratch** each time. This leads to an O(T^2) complexity in trajectory length T. In contrast, OEC requires only a single switch, enabling reuse of the KV cache after switching, only requiring recomputing the KV cache a single time and yielding an O(T) complexity. Our early experiments showed that that this practically made experiments with ordinary DAgger infeasible, even at the 7B scale
> 2. *Compatibility with rejection sampling*:  A key goal of our setting is to leverage verifier-based filtering (e.g., unit tests). Full DAgger-style querying only provides single-step expert labels and does not produce complete trajectories, **making it difficult to apply rejection sampling**. In contrast, OEC produces full expert-completed trajectories, enabling the use of verifiable rewards.
>
> > (2) The OEC data consist of erroneous trajectories generated by the student model along with corrections provided by the expert model, so the dataset may still contain some incorrect instances. Could the authors provide a more detailed analysis of the proportion of expert model interventions in the data?
>
> Every OEC trajectory includes an intervention from the expert. Moreover, we mask out the student (on-policy) portion and train only on the expert-corrected suffix of each trajectory (Section 3.2), except for the ablation in Table 2 where we consider training on the on-policy portions, too. Our experiments find that combining OEC data with standard behavioral cloning (BC) yields the strongest performance. Figure 3 shows results in which we ablate a varying amount of OEC and BC data mixed together.

---

> > ### Author Rebuttal · Reviewer_RnQB · 2026-04-03
> >
> > Thank you to the authors for addressing my questions, and I have also seen the comparison with the new Agent-RLVR method. I will maintain my previous positive rating.

---

### Decision · Program_Chairs · 2026-04-30

**Decision:**

Accept (regular)

**Comment:**

This paper proposes an adaptation of the DAgger algorithm used to address covariate shift in imitation learning to the LLM framework. In particular, rollouts which are started using the current LLM policy are passed to the expert (a higher capacity LLM) to be completed and added to the data set, thus filling in expert actions on the states where the policy might drift to. All reviewers were overall positive about this paper and recommended acceptance (3 weak accept, 1 accept), hence I recommend accept.